# Aerial survey estimates of polar bears and their tracks in the Chukchi Sea

**Paul B. Conn**[ID]**[1]***, **Vladimir I. Chernook**[2], **Erin E. Moreland**[1], **Irina S. Trukhanova**[ID]**[3]**, **Eric V. Regehr**[4,5], **Alexander N. Vasiliev**[2], **Ryan R. Wilson**[4], **Stanislav E. Belikov**[6], **Peter L. Boveng**[1]

**1** Alaska Fisheries Science Center, National Marine Fisheries Service, National Oceanic and Atmospheric Administration, Seattle, Washington, United States of America, **2** Ecological Center, Autonomous Non-Commercial Organization, Saint-Petersburg, Russia, **3** North Pacific Wildlife Consulting, LLC, Seattle, Washington, United States of America, **4** Marine Mammals Management, United States Fish and Wildlife Service, Anchorage, Alaska, United States of America, **5** Applied Physics Laboratory, Polar Science Center, University of Washington, Seattle, Washington, United States of America, **6** All-Russian Research Institute for Nature Protection (Federal State Budgetary Institution), Moscow, Russia

* paul.conn@noaa.gov

**Data Availability Statement:** Code and data to recreate the analyses are located at https://github.com/pconn/ChukchiPolarBear and permanently

## Abstract

Polar bears are of international conservation concern due to climate change but are difficult to study because of low densities and an expansive, circumpolar distribution. In a collaborative U.S.-Russian effort in spring of 2016, we used aerial surveys to detect and estimate the abundance of polar bears on sea ice in the Chukchi Sea. Our surveys used a combination of thermal imagery, digital photography, and human observations. Using spatio-temporal statistical models that related bear and track densities to physiographic and biological covariates (e.g., sea ice extent, resource selection functions derived from satellite tags), we predicted abundance and spatial distribution throughout our study area. Estimates of 2016 abundance ($\hat{N}^*$) ranged from 3,435 (95% CI: 2,300-5,131) to 5,444 (95% CI: 3,636-8,152) depending on the proportion of bears assumed to be missed on the transect line during Russian surveys ($g(0)$). Our point estimates are larger than, but of similar magnitude to, a recent estimate for the period 2008-2016 ($\hat{N}^* = 2,937$; 95% CI 1,522-5,944) derived from an integrated population model applied to a slightly smaller area. Although a number of factors (e.g., equipment issues, differing platforms, low sample sizes, size of the study area relative to sampling effort) required us to make a number of assumptions to generate estimates, it establishes a useful lower bound for abundance, and suggests high spring polar bear densities on sea ice in Russian waters south of Wrangell Island. With future improvements, we suggest that springtime aerial surveys may represent a plausible avenue for studying abundance and distribution of polar bears and their prey over large, remote areas.

## Introduction

Polar bears (*Ursus maritimus*) are distributed throughout the circumpolar Arctic in 19 partially discrete subpopulations [1]. In 2008, the species was listed as threatened under the U.S.

archived on Zenodo at doi: 10.5281/zenodo.
4708335.

**Funding:** Funding for surveys was provided
primarily by the U.S. National Oceanic and
Atmospheric Administration (NOAA) and the U.S.
Fish & Wildlife Service (USFWS). These funders
provided support in the form of salaries for authors
[PC, EM, ER, RW, and PB], but did not have any
additional role in the study design, data collection
and analysis, decision to publish, or preparation of
the manuscript. The specific roles of these authors
are articulated in the 'author contributions' section.
Support for Russian surveys was provided by
NOAA through the North Pacific Wildlife
Consulting, LLC (http://www.northpacificwildlife.
com/). Portions of the analysis were supported by
joint subaward NA17NMF4720289, project 1813,
from the North Pacific Research Board and The
Prince William Sound Oil Spill Recovery Institute
(https://www.nprb.org/core-program/about-the-
program/; PL, ER, IT, EM, and PC were principal or
co-investigators). Additional support for data
processing and survey logistics on the Russian
side was provided by USFWS, the RPO Marine
Mammal Council (https://marmam.ru/en/) and
WWF Russia (https://wwf.ru/en/about/) in funding
agreements with VC. External funders (e.g., NPRB,
WWF Russia) had no role in study design, data
collection and analysis, decision to publish, or
preparation of the manuscript.

**Competing interests:** "The primary funders (U.S.
National Oceanic and Atmospheric Administration;
U.S. Fish & Wildlife Service) provided salaries for
authors [PC, EM, ER, RW, and PB] but did not have
any additional role in study design, data collection
and analysis, decision to publish, or preparation of
the manuscript. This does not alter our adherence
to PLOS ONE policies on sharing data and
materials."

Endangered Species Act due to loss of sea-ice habitat resulting from climate change [2]. Currently, population trends of the 19 subpopulations is variable due to a number of factors, including geographic variation in sea-ice conditions and ecosystem function [3, 4]. Subpopulation-specific data are required to understand the effects of climate change and inform localized conservation solutions, including management of subsistence harvests that provide nutritional, cultural, and economic benefits to Indigenous people [5, 6]. However, accurate and timely population data are difficult and expensive to collect because polar bears inhabit remote regions in low densities. For instance, previous estimates of springtime (April) density obtained from mark-recapture analysis ranged from 0.001-0.01 (mean 0.004) bears/km$^2$ in the Canadian Arctic [7], and 0.003 bears/km$^2$ for the Chukchi Sea [8]. Aerial survey estimates of polar bear densities are often conducted in late summer and early fall when polar bears are in higher concentrations because of reduced sea ice; densities at this time of year have ranged from 0.001 bears/km$^2$ in the Barents Sea [9] where there is still substantial sea ice, to 0.02 bears/km$^2$ in Southern Hudson Bay when sea ice has melted completely and bears are confined to land [10].

The Chukchi Sea (CS) polar bear subpopulation (also referred to as the Alaska-Chukotka population, with different boundaries) ranges widely on sea ice of the northern Bering, Chukchi, and East Siberian seas [11] and is managed under a bilateral treaty between the U.S. and Russia [1]. The first estimates of Alaska-Chukotka polar bear abundance (2,000-5,000 bears) were published in the early 1990s, based on both expert opinion and the number of maternal dens counted on Wrangel Island, Russia (inflated to account for breeding females comprising 8-10% of the population) [12, 13]. Live-capture research in the U.S. between 2008 and 2016 suggested that CS bears displayed good nutritional condition [14, 15] and productive demography [8] despite sea-ice loss. An integrated population model fitted to these data and extrapolated to the CS region produced an estimate of 2,937 (95% CI 1522-5944) bears. Capture research is unlikely to be performed annually in the future, however, because of high costs and human safety concerns due to progressive deterioration of spring sea-ice conditions. Alternative study methods are needed to understand the mechanisms through which sea-ice loss is affecting CS bears and to periodically update population data according to a harvest management framework adopted in 2018 [16].

Aerial surveys have been used to estimate polar bear subpopulations in a number of regions, and are typically conducted in late summer and early fall when there is less sea ice and bears are most concentrated [9, 10, 17]. However, another possible approach is to conduct aerial surveys during the spring; although bears will be spread over a larger area, this approach allows one to study their distribution over the sea ice, and to simultaneously study the distribution of seal populations (the primary prey of polar bears) when they are engaged in pupping and molting and are therefore most available to be sampled [18]. Conducting surveys in spring also allows for the potential of instrument-based approaches in which infrared cameras and coordinated digital color photography can be used to detect the warm bodies of animals on sea ice and classify species. Such surveys have proven extremely effective for estimating ice-associated pinniped abundance [19–21], but have yet to be applied to polar bears (except to detect dens [22]).

To better understand the distribution and abundance of seals and polar bears in the CS region, we conducted a comprehensive, instrument-based aerial survey in April and May of 2016. Although our survey generated count data for multiple species (including seals), here we focus on polar bears, including direct observations of animals and observations of tracks in snow accumulated on sea ice. We provide an overview of the survey area and study design, including technical specifications of survey equipment and data processing protocols, and describe novel statistical methods. We then use our methods to estimate the distribution and abundance of polar bears in the CS region.

## Materials and methods

Aerial surveys were conducted under National Marine Fisheries Service permit Permit No. 19309 and U.S. Fish & Wildlife Service Permit No. MA212570-1.

### Study area

We conducted aerial surveys over the Chukchi Sea between April 7 and May 31, 2016. Our study area was bounded to the south by the Bering Strait, to the north by the U.S. and Russian Exclusive Economic Zone boundaries, to the east by the 156˚ W line, and to the west by a line extending north from Chaunskaya Bay, Russia (Fig 1). We divided our study area into grid cells that were approximately 25 km by 25 km, based on a variant of the Lambert azimuthal equal-area projection. We chose this scale because it corresponded to the resolution of sea ice imagery downloaded from the National Snow & Ice Data Center (NSIDC; see *Explanatory covariates*, below) and for consistency with previous analyses of ice-associated seals in the Bering Sea [21]. Removing portions of grid cells that included land, the total area of our study area was 798,049 km$^2$. Our study area included all marine habitat within this region, including open water and areas covered by sea ice (though we set polar bear abundance to zero in cells with no ice; see *Models and model fitting*). U.S. survey flights were conducted out of Kotzebue, Alaska, U.S.A. and Utqiaġvik, Alaska, U.S.A., whereas Russian flights operated out of Pevek, Chukotka, Russia, and Provideniya, Chukotka, Russia.

Ambient conditions varied considerably during the survey period. Weather at Utqiaġvik and in the northeast quadrant of the study area started off cold, clear, and windy, with temperatures often < −10˚C with average windspeeds near 10 m/s. This pattern changed to one with warmer temperatures (up to 7˚C by the end of May), with less wind, but with persistent fog that often hampered survey efforts during the second half of our surveys. Weather in Kotzebue, Provideniya, and in the southeast quadrant of our survey area was more benign, with average daytime highs near 0˚C in April, climbing to near 16˚C by the end of May. In Pevek, wind was calm but temperatures cold (-18 to -5˚C) up until May 6, when temperatures rose slowly to day time highs near 0˚C by mid-May. Most of our study area was covered in sea ice at the beginning of our surveys, but by the end of May there were large ice free areas north of the Bering Strait and near coastal Alaska (S1 Video).

### Aerial survey platform and protocols

We used two aircraft with different observation platforms to conduct aerial surveys. In U.S. airspace, surveys were flown in a King Air A90, crewed with a pilot, a copilot, and two scientists. The King Air A90 was configured with three downward facing, cooled long wavelength infrared (LWIR) cameras (FLIR A6750sc SLS) with 25 mm lenses mounted in the bellyport. An uncooled LWIR microbolometer (FLIR A645sc) was used as a backup camera for a portion of the survey. The thermal cameras had a viewing angle of 24.9˚ and temperature sensitivity of 0.03˚C. Each of these thermal cameras was paired with a machine vision 29 megapixel camera (Prosilica GT6600c) fitted with a 100 mm lens which provided matching color images. Camera pairs were mounted in an array with the center pair in a nadir position, and side pairs mounted at 25.5˚ angles inward, providing an overall field of view of 76˚ for the thermal cameras. Flying at a target altitude of 305 m, the ground resolution of the thermal cameras was 20-23 cm/pixel with a swath width of 470 m. The ground resolution of the color cameras was 1.71-2.13 cm/pixel with a footprint that closely matched (but did not completely cover) the thermal footprint of each camera. Image collection between the color and thermal cameras was synchronized and images were collected continuously at a rate of approximately 2 frames per second (fps), providing forward moving overlap of images when flying at 222-259 km/hr.

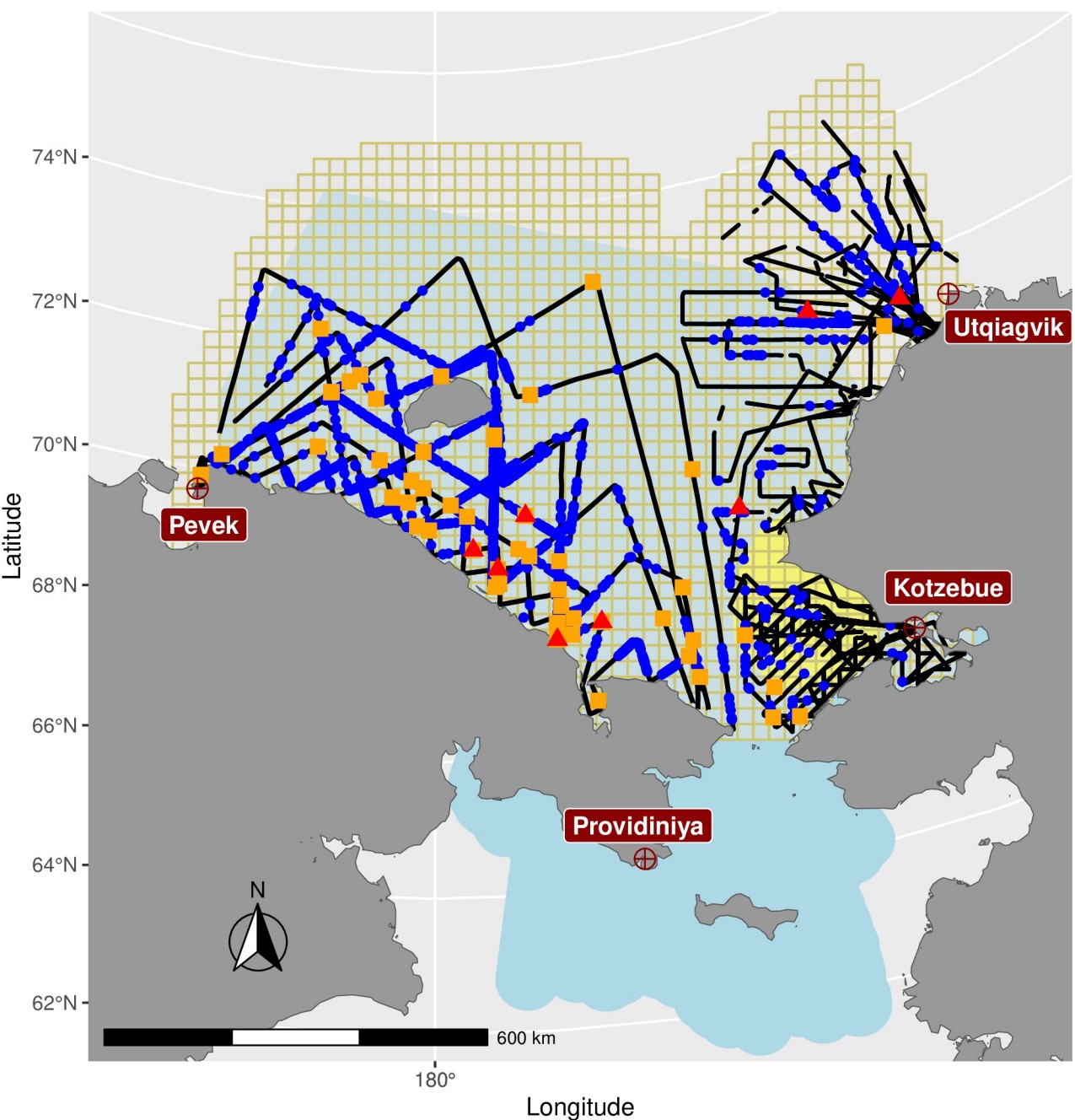

**Fig 1. Chukchi Sea study area.** A map of the study area used in 2016 aerial surveys for seals and polar bears. Black lines represent aerial survey tracks, while small blue circles represent locations of bear tracks. Breaks in transect lines represent times where survey crew went "off effort" because of dense fog. Red triangles represent thermal detections of polar bear groups, and orange squares represent additional groups seen by human observers (U.S. and Russia) or in post hoc examination of photographs (Russia only). The ≈625km² grid cells used for abundance estimation appear in the background (beige lines). Land masses (gray) include Alaska, U.S.A. to the east and Russia, to the west. The blue shading represents the area associated with the Chukchi subpopulation of polar bears as determined by the polar bear specialist working group (PBSG), while yellow shading shows the area where a long term capture-recapture study of polar bears [8] was previously conducted. For an animation depicting survey effort and observations overlayed on sea ice as a function of survey day, see S1 Video.

In addition to collecting automated color and thermal imagery, the U.S. survey team recorded visual observations of bears that occurred outside the range of the thermal swath.

In Russia, surveys were flown using an AN-26 Arctica aircraft, with a crew of 7 (pilot, copilot, infrared (IR) scanner operator, and four visual observers). The aircraft was equipped with a Malachite-M IR scanner with a wide viewing angle of 88˚ and temperature sensitivity of 0.2˚C. An Optris PI 450 thermal camera was used as a backup. Flying at an average speed of 296 km/hr and a target altitude of 250 m, the IR ground footprint was 483 m and was completely covered by three digital color 36 megapixel cameras with 50 mm lenses. The IR image resolution at the target altitude was ≈11 cm/pixel and the resolution of color cameras was 2.27 cm/pixel. Color cameras could be triggered by any of the following events: 1) a real-time IR hot spot detection algorithm triggered the cameras; 2) IR-scanner operator (a technician monitoring the IR-video stream in real-time) detected an anomaly (possible hot spot, not strong enough to be picked up by the detection algorithm) on the screen and manually triggered the color cameras; or 3) one of the visual observers saw an object of interest on ice (e.g., animal, tracks, kill site, water access hole) and manually triggered the color cameras. Additionally, the color cameras collected photographs at a set interval of 20 sec (for May flights only). In addition to imagery, four visual observers (two on each side of the aircraft) made continuous observations through bubble windows and used hand-held inclinometers to determine the angle at which an object of interest was observed in order to estimate detection distance. Visual observations included areas both within, and extending beyond, the thermal swath boundary.

Aerial surveys of wildlife often use design-based statistical inference to estimate abundance. This approach requires survey planners to define a sampling frame of all possible transects, and to sample amongst those (often using systematic random sampling [23]) prior to conducting the survey. By contrast, model-based estimation, including modern density-surface models applied to data from line transect surveys [24] does not suffer from this requirement (though randomization guards against subjective decisions that have potential to bias survey results through preferential sampling [25, 26]). Model-based estimation has the key ramification that transect placement does not need to be allocated prior to the survey, permitting flexibility in decisions about when and where to survey, which is invaluable for modifying surveys when weather (e.g., fog) precludes surveying in certain areas.

A previous study examining alternative transect placement strategies for aerial surveys in the eastern Chukchi Sea [27] suggested reasonable precision and lack of bias when applying model-based estimation procedures to simulated polar bear count data. In that study, spreading effort out evenly over space resulted in slightly improved inference compared to stratified designs. This result was similar to what has been observed when fitting spatial models to environmental pollutant data: space-filling designs (in which sampling effort is spread evenly over space) tend to be optimal [28]. Given this finding, our primary philosophy when making and altering flight plans (as sea ice conditions and weather changed, for instance) was to spread out sampling effort over time and space. We avoided surveying grid cells that were 100% open water, but otherwise attempted to structure transects to sample representative habitat within grid cells that did have ice.

U.S. and Russian survey protocols differed substantially, mostly owing to the constraints imposed by the survey platforms used. In the U.S., pre-survey flight planning supposed 21 flights totalling 20,950 km of "on effort" data collection (777-1280 km per flight). Planned flights consisted of 2.6-4.3 hrs of survey effort, centered on solar noon to maximize the number of seals that would be encountered [18]. However, variable weather conditions resulted in opportunistic survey effort and transects that varied considerably from these targets (see Results). The Russian survey team initially planned to fly 8 transects covering 13,000 km of transect line over 43 "on effort" flight hours (roughly 1625 km and 5.4 hours per flight).

To avoid potential for bias due to preferential sampling [25, 26] crews of both aircraft were instructed to avoid fine scale targeting of ice habitat (e.g. following leads) or areas of high seal density when making and altering flight plans as sea ice and weather conditions changed. Owing to less flexibility in modifying transects while in flight, the Russian aircraft largely followed predetermined flight lines, while U.S. aircraft frequently made adjustments to sample areas that had not previously been sampled, or to avoid areas where visibility was poor (Fig 1).

## Data and data processing

**Hot spot detection & post hoc searching of photographs.** After surveys were completed, U.S. analysts used a general outlier algorithm to detect warm heat signatures of animals on sea ice and matched IR detections with paired color photographs (Fig 2). Due to high false-positive rates of the algorithm, manual review of detections was implemented to eliminate hot spots caused by features other than animals (e.g., melt pools, dirty ice). Since the infrared scanner on the Russian aircraft appeared not to detect many bears, Russian analysts opted to manually examine each color photograph for presence of polar bears. For each bear detected in Russian surveys, analysts calculated the distance of the bear from the transect line using standard trigonometry.

**Polar bear tracks.** Polar bear tracks were frequently encountered in our surveys, and we summarized the frequency of polar bear tracks as a potential correlate for polar bear density. For U.S. flights, we reviewed every 20th color image from our port-side camera and recorded whether or not it included tracks. In Russian flights, we summarized the number of tracks recorded by human observers in each surveyed grid cell.

**Polar bear detection trials.** Previous studies of ice-associated seals documented high (e.g., $p = 0.96$) detection probabilities using the same thermal detection algorithm applied to the U.S. data set [21]. However, aerial thermal imagery was not available to evaluate the accuracy of the algorithm for detecting polar bears prior to this survey effort. Despite the high emissivity of polar bear hair [29], the thermal signature of bears was likely to differ from that of seals based on the texture of the bears fur and the shape of the animal [30]. To determine the detection probability of polar bears in U.S. survey flights, we conducted experimental

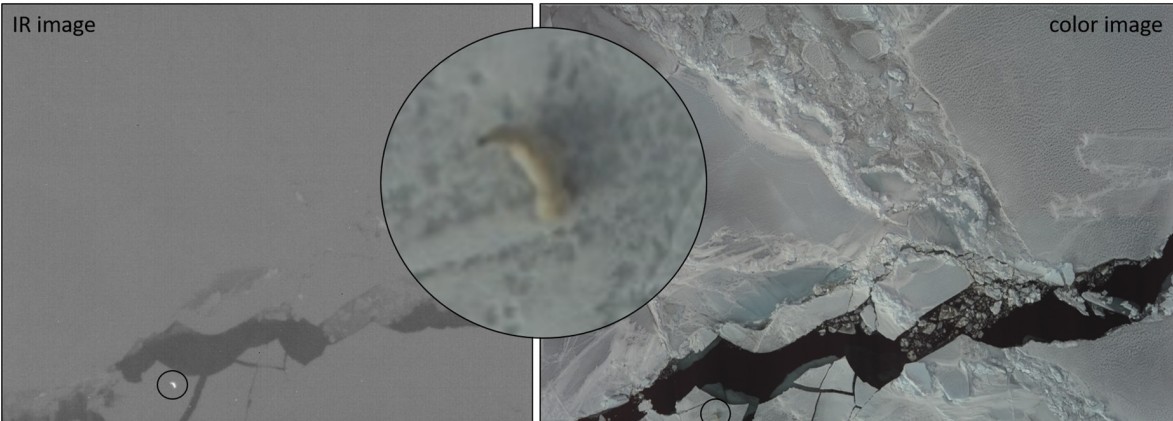

**Fig 2. Paired infrared and color imagery of a polar bear.** Color (right) and IR (left) imagery collected from 300 m during the 2016 aerial surveys from the U.S. platform containing a polar bear (zoomed inset). The IR image collected with the long wavelength infrared cooled camera (FLIR A6750sc SLS) provides a visual representation of apparent temperature in greyscale where darker shades are cool and lighter shades are warm. The paired color image, collected with the Prosilica GT6600c fitted with a 100 mm Zeiss lens, confirms that the heat signature detected in the IR image is from a polar bear.

flyovers. Specifically, each time a polar bear was detected from the air by a human observer, we continued on transect past the bear and then went "off effort" and conducted focused fly overs while recording thermal imagery and digital photography. We also conducted flights over polar bears congregating near whale carcasses close to the village of Utqiaġvik, Alaska. In each case, a technician manually reviewed digital photographs with time stamps associated with our fly-overs in an effort to locate polar bears. In some cases, technicians had also participated in survey flights, and in some cases had not. Such polar bear detections became trials with which to estimate detection probability of our thermal detection algorithm. In total, we documented 12 unique polar bears in flyovers from the manual photograph review. Of these, we detected 8 with our thermal detection algorithm and manual hot spot review, for an apparent detection probability of $\hat{p}_{us} = 0.67$ (SE 0.14).

The Russian aircraft was considerably less maneuverable than the U.S. King Air so focused flyovers were not tenable. Further, the Malachite-M thermal scanner appeared to have trouble detecting bears. No bears were detected in temperatures $< -5°C$ (temperatures typical of April survey flights). Even when temperatures were $> -5°C$, the thermal scanner detected just 1 out of 9 visually detected bears. Due to the unreliability of the Russian thermal sensor for bear detection, we relied on distance sampling to estimate detection probability within the Russian portion of the survey area (see *Polar bear counts*, below).

**Explanatory covariates.** We assembled a number of environmental and landscape covariates that we thought might help to predict the density of polar bears and their tracks, including.

- dist_land: distance from the centroid of each grid cell to the closest point of land (Alaska or Russia);

- ice: daily remotely sensed sea ice concentration values, as obtained from the NSIDC, Boulder, Colorado, USA;

- RSF: an estimate of the relative density of space use obtained by fitting resource selection functions (RSFs) to data from fixes of satellite-collared female polar bears. We followed the same procedure as Wilson et al. (2016) [11], updating their RSFs with environmental data from 2016 (see S1 Appendix for further details and representative maps);

- easting: standardized longitude-like values in projected geographic space;

- northing: standardized latitude-like values in projected geographic space;

- Water99: a binary indicator for whether or not sea ice concentrations were less than 1%.

The ice covariate was often missing or unreliable in grid cells that bordered mainland Alaska or Russia. For these cells, values of ice were interpolated via kriging using the auto-Krige function in the automap library [31] within the R programming environment [32]. Absolute correlations between explanatory covariates were all <0.5, with exception of northing and dist_land, which had a correlation of 0.75.

We included dist_land because seal densities (the primary prey of polar bears) are often highest close to land [18], and also because maternal polar bear dens are often located on land [33] with high concentrations on Wrangel Island, Russia [34] and along the northern Alaska coast [35]. Since mothers and cubs emerge from dens in late winter and early spring (March-early April) we suspected there may be higher densities of bears along coastlines. We included ice since it has repeatedly been demonstrated to be an important determinant of polar bear habitat selection [36–38]. Similarly, we included Water99 as a way to restrict polar bear use of habitat to those grid cells with >1% ice. Although bears can swim long distances, it was

impossible to detect bears in the water and the proportion swimming at any one time is thought to be extremely low. The RSF distribution was a measure of habitat use developed from adult females; if habitat preferences of these bears mirror that of the population, we expected it would be a reasonable correlate for overall polar bear densities. Although the easting and northing covariates have little ecological meaning, we included them in models for polar bear counts because they enabled us to model coarse-grained spatial autocorrelation (clustering) in bear densities, as common in geostatistics and spatio-temporal statistical models [39]. Previous research [38] found that polar bear resource selection can also depend on additional covariates such as proportion of landfast ice, ocean depth, variability of sea ice concentration, and average spring-fall chlorophyll concentration. Although we did not directly include these covariates in our models, most were implicitly included in our RSF covariate (see S1 Appendix). For a description of which covariates were used in models for polar bear tracks and count data, see *Models and model fitting*, below.

## Statistical analysis

We fitted several spatio-temporal statistical models to polar bear track and count data. In each case, we modeled track and count data simultaneously and aggregated them into discrete grid cells $\approx 25 \times 25$ km$^2$ (Fig 1) prior to analysis.

**Polar bear tracks.** In order to derive a single covariate describing density of polar bear tracks, we jointly modeled U.S. and Russian track data. This was challenging since these data were summarized on different scales, requiring a common currency to describe both binary detection data in photographs (U.S.) and total number of tracks observed by human observers (Russia). We start by making the simplifying assumption that the distribution of tracks are governed by a "blocky" spatial Poisson point process with rate parameter $Z_{s,t}$ that is assumed constant for surveys of grid cell $s$ at time $t$. According to this model, the probability of observing $Y = y$ tracks in an arbitrary area $B$ is

$$\Pr(Y = y) = \frac{(BZ_{s,t})^y}{y!} \exp\left(-BZ_{s,t}\right).$$

Technically this model assumes spatial independence of tracks, which is surely not satisfied since tracks are often continuous circuitous features left by a relatively small number of animals. However, we were willing to accept some level of model misspecification and attendant overdispersion in order to come up with a joint predictive covariate, because neither absolute track density nor accurate estimation of its variance are needed to construct a track density index.

Using this formulation for track density, and setting $B = 0.012$ km$^2$ (the average ground footprint of an examined photograph), the probability of observing one or more tracks in a U.S. photograph is

$$\phi_{s,t} = \Pr(Y > 0) = 1 - \exp\left(-0.012Z_{s,t}\right).$$

We then modeled the number of U.S. photographs with polar bear tracks in cell $s$ at time $t$ ($T_{s,t,us}$) as

$$T_{s,t,us}|P_{s,t} \sim \text{Binomial}(T_{s,t,us}; P_{s,t}, \phi_{s,t}),$$

where $P_{s,t}$ gives the number of photographs examined in cell $s$ at time $t$.

For Russian track detections, we modeled the number of tracks recorded by human observers in cell $s$ at time $t$ ($T_{s,t,rus}$) as

$$T_{s,t,rus} \sim \text{Poisson}(L_{s,t}\eta Z_{s,t}),\qquad(1)$$

where $L_{s,t}$ gives the length (km) of the transect flown over cell $s$ at time $t$. We included an additional scaling parameter, $\eta$, in Eq (1) because (i) there was uncertainty about the effective strip width for tracks since distances were not recorded, and (ii) we were uncertain about how separate tracks were delineated. For instance, it might be possible for a single set of polar bear tracks to encompass a large area. This scaling parameter thus helps to spatially align U.S. and Russian datasets.

We imparted spatio-temporal variation in $Z_{s,t}$ using the construction

$$\log(Z_{s,t}) = \mathbf{x}'_{s,t}\boldsymbol{\beta},$$

where $\mathbf{x}_{s,t}$ is a column vector of predictive covariates for cell $s$ at time $t$ and $\boldsymbol{\beta}$ is vector of regression coefficients. For details on specific models fit, see *Models and model fitting*.

**Polar bear counts.** When the goal of a survey is to estimate absolute abundance, several avenues exist for modeling spatio-temporal variation in animal counts [40]. However, when data are sparse (as in the case of polar bear counts), considerable stability is introduced by assuming a population closure assumption, whereby abundance is assumed constant in a study area but animals are allowed to redistribute themselves as conditions change (e.g., as ice melts) [40, 41].

We adopted such a dynamic redistribution model for polar bear abundance, but focus on the number of distinct polar bear groups (typically 1-3 bears) and employ a separate model for group size (see below). In particular, we assumed a fixed number of $N$ polar bear groups were present in the study area on each day of the survey, and that the number of bear groups in each grid cell $s$ ($s = 1, 2, \ldots, S$) at time $t$ ($t = 1, 2, \ldots, T$), $N_{s,t}$, could be described as

$$N_{s,t} = N\pi_{s,t}.$$

Here, $\pi_{s,t}$ gives the probability a polar bear group is in grid cell $s$ at time $t$ (note that $\sum_s \pi_{s,t} = 1.0$). We specify a multinomial link function for $\pi_{st}$, such that

$$\pi_{s,t} = A_s \exp(v_{s,t})\Big/\sum_s A_s \exp v_{s,t},$$

where $v_{s,t}$ represents a linear predictor (without an intercept), and $A_s$ denotes the proportion of grid cell $s$ that is composed of saltwater habitat (sea ice and water, but not land). Predictive covariates are introduced within a linear modeling construct, such that

$$v_{s,t} = \mathbf{y}'_{s,t}\boldsymbol{\alpha},$$

where $\mathbf{y}_{s,t}$ is a vector of predictive covariates for grid cell $s$ at time $t$ and $\boldsymbol{\alpha}$ is a vector of regression parameters.

We modeled polar bear group counts with a Poisson distribution, such that

$$C_{s,t}|N_{s,t}, p_{s,t} \sim \text{Poisson}(N_{s,t}p_{s,t}).$$

where $p_{s,t} = \theta_{s,t} a_{s,t}$ is a compound detection probability that represents a number of components, including (i) imperfect detection of bears on ice ($\theta_{s,t}$), and (ii) incomplete coverage of aerial surveys over grid cell $s$, such that $a_{s,t}$ represents the proportion of salt water habitat in grid cell $s$ that is surveyed during transects on day $t$. For U.S. surveys, we limited thermal detections to those where species were confirmed on photographs, so we calculated $a_{s,t}$ using

the total ground footprint of photographs. For Russian surveys, we calculated $a_{s,t} = L_{s,t} * 1.2/A_{s,t}$, where $L_{s,t}$ was the length of transect flown in cell $s$ at time $t$, and the 1.2 represents a fixed width of 0.6 km on either side of the aircraft (which is motivated by the distribution of observed polar bear distances; see below). We set $\theta_{s,t} = p_{us}$ for U.S. surveys and $\theta_{s,t} = p_{rus}$ for Russian surveys, which are detection probability of bears for U.S. and Russian surveys, respectively.

For U.S. surveys, information about detection probability, $p_{us}$ was obtained from experimental flyovers of polar bears. In particular, we assumed that the number of detected bears in experimental flyovers (D) was binomially distributed with an index equal to the number of first flyovers (n = 12) and success probability $p_{us}$:

$$D \sim \text{Binomial}(n, p_{us}).$$

For Russian surveys, we used distance sampling ideas to model detection probability. In particular, we write

$$p_{rus} = g(0) \int_0^w \frac{h(x)}{wh(0)}\ dx,$$

where $h(x)$ is a probability density function describing the distribution of observed distances (using either in situ observations of distance made using an inclinometer or post hoc examination of photographs) from the transect line, and $g(0)$ gives the probability of detection on the transect line. We fitted a number of detection functions to distance data, and determined that the half normal distribution adequately characterizes the decline in detections as a function of distance (S2 Appendix). Note that our comparison of detection functions included an option for a uniform-half normal mixture, where the uniform distribution applied to distances of 350 m from the aircraft and the normal distribution applied out to the truncation distance of $w$ = 600 m. Since photographs were only obtained up to a fixed distance from the aircraft (median = 350 m) depending on altitude, one might expect this mixture detection function to better apply to our data (photographs + human observers), but the half-normal was favored by AIC ($\Delta$AIC = 1.6). We thus used it as the functional form of $h$, treating $\log(\sigma)$ (the log of the half normal standard deviation) as an unknown parameter.

The primary challenge with this approach is that data were not gathered in a manner amenable to estimating $g(0)$, the proportion of animals that are seen on the transect line (i.e., underneath the aircraft). Although $g(0)$ from other polar bear surveys are available (and most are close to 1.0; S2 Appendix), most of these studies used helicopters traveling at substantially slower speeds and lower altitudes than the fixed wing aircraft used in Russian surveys. Many of these surveys were also conducted in fall over snow-free land where bears are more visible (S2 Appendix). For these reasons, in situ detection probability of human observers was potentially much lower than reported in the literature. On the other hand, we also incorporated detections of bears from post hoc examinations of photographs into our distance sampling analysis which markedly increased sample size and raised $g(0)$ relative to its value had we just used in situ observations. Given these confounding factors, it is difficult to make an objective determination of a likely value for $g(0)$. We thus considered a range of values including $g(0)$ = 0.6, 0.8, and 1.0.

If we limited analysis to counts made by thermal cameras while "on effort" for U.S. surveys we would be limited to $n$ = 3 sightings. However, we also had access to an additional $n$ = 5 detections of bear groups that were detected visually while on transects (often beyond the swath of our thermal scanners). However, we did not collect distance measurements in U.S. surveys, so there was no way to calculate effective strip width [42]. As such, we did not know

the effective area surveyed nor have a good idea of detection probability for this class of observations. Nevertheless, these data are clearly informative about the locations and habitats where bears were detected and we wished to use them to improve estimation of density-habitat relationships. To incorporate these data, we modeled these auxiliary bear group counts ($U_{s,t}$) as

$$U_{s,t} \sim \text{Poisson}(\xi_{us} N \pi_{s,t} p_{s,t}),$$

where $\xi_{us}$ is an estimated parameter representing the relative detectability of auxiliary counts relative to thermal detections in U.S. surveys. We assumed this ratio remained constant over the course of the survey.

Thus far, we have described models for the abundance of polar bear groups, as opposed to total abundance. This is because we treated groups as the primary sampling unit when analyzing transect data [42]. We estimated absolute abundance, $N^*$, as $N^* = \mu_g N$, where $\mu_g$ is a parameter representing mean group size. We modeled observed group size ($g_i$) for each of $i = 1, 2, \cdots, 49$ groups as a realization from a zero-truncated Poisson process; that is,

$$(g_i - 1) \sim \text{Poisson}(\mu_g - 1).$$

**Models and model fitting.** We fitted three different models to count and detection data, corresponding to three assumed values for $g(0)$ in Russian surveys: 0.6, 0.8, or 1.0. The model for polar bear tracks was relatively simple and used simple polynomial regression such that $\mathbf{x}'_{s,t}$ included an intercept, linear and quadratic effects of distance from land, linear and quadratic effects of sea ice concentration, and a linear effect of the resource selection function value (RSF) for a given cell.

Models for latent polar bear abundance were considerably more complex, as we wanted the data to "speak for themselves" by allowing smooth effects of covariates similar to the generalized additive modeling (GAM) framework [43] commonly employed in modern density surface modeling of species distributions [24]. To that end, we adopted a penalized spline formulation for smooth effects [44]. First, we used the `mgcv` package [43] in the R programming environment [45] to construct cubic smoothing bases and penalty matrices. We set the maximum basis dimension for each covariate to six to reduce dimensionality; each smooth effect thus included six parameters. Following the approach implemented in the 'jagam' function in the `mgcv` package [46]. we implemented penalties by imposing prior distributions on the regression coefficients, $\boldsymbol{\alpha}$. In particular,

$$\boldsymbol{\alpha} \sim \text{Multivariate normal}(\mathbf{0}, \boldsymbol{\Sigma}),$$

where $\boldsymbol{\Sigma}$ is a block diagonal matrix with block entries given by $\lambda_i \mathbf{S}_i$. Here, $\mathbf{S}_i$ gives a penalty matrix output from `mgcv` and $\lambda_i$ is a parameter that controls the degree of smoothing. As in the 'jagam' function, we assigned Gamma prior distributions to the $\lambda_i$ parameters:

$$\lambda_i \sim \text{Gamma}(0.05, 0.005).$$

In bear count models, we included latent polar bear track intensity ($Z_{s,t}$) as an additional linear fixed effect. Inference about $N$ thus properly accounts for uncertainty in the relative frequency of bear tracks. In order to prevent our model from predicting polar bears or tracks in grid cells without (or with very little) sea ice, we included the covariate `Water99` in all models fitted, and set the corresponding regression coefficient equal to -50.0. For reference, all but two sightings of polar bears occurred in grid cells with sea ice concentrations >80% (although one bear was observed in a cell with ≈3% ice).

In order to propagate uncertainty from track, count, detection probability, and group size models into final abundance estimates, we based statistical inference for polar bear abundance on the joint marginal likelihood

$$L = \int_{\alpha} \int_{\lambda} L_t L_c L_u L_p L_g P_{\alpha} P_{\lambda} \; d\boldsymbol{\alpha} \; d\lambda,$$

where $L_t$ represents the likelihoods for U.S. and Russian polar bear tracks, $L_c$ denotes Poisson likelihoods for instrument-based and distance-sampling counts (for U.S. and Russian surveys, respectively), $L_u$ denotes a Poisson likelihood for U.S. human observer counts, $L_p$ denotes binomial and half normal likelihoods for U.S. flyovers and Russian distance data, $L_g$ is the zero-truncated Poisson likelihood for group size, and $P_{\alpha}$ and $P_{\lambda}$ give prior distributions for spline parameters $\boldsymbol{\alpha}$ and $\lambda$, respectively. We treated spline parameters as random effects, and integrated them out of the likelihood using the Laplace approximation capability in Template Model Builder [47] (see *Software* below).

Log and logit link functions were used on bounded parameters to allow unbounded optimization. Point estimates and Hessian-based standard errors were used to construct 95% log-based confidence intervals [42, 48] for total abundance.

**Goodness-of-fit.** To examine the fit of track and bear count submodels, we examined distributions of randomized quantile residuals (RQRs) [49]. Such residuals are especially useful for discrete random variables, where it can be difficult to assess lack-of-fit visually because of "clumping" at fixed values. For an arbitrary datum $y_i$, randomized quantile residuals were simulated as

$$R_i = F(y_i|\hat{\mu}_i) + u_i f(y_i|\hat{\mu}_i),$$

where $F()$ gives a cumulative distribution function (CDF) with mean $\hat{\mu}_i$, $u_i$ is a uniform random deviate, and $f()$ is a probability mass function (PMF). For example, for a Poisson count model, $y_i$ represents an observed count, $\hat{\mu}_i$ represents a predicted count, and $F()$ and $f()$ give the Poisson CDF and PMF, respectively. If a model fits the data well, RQRs computed in this way should be uniformly distributed on (0,1). To assess uniformity, we calculated chi-squared test statistics with 10 equally sized bins for each count or track submodel.

## Software

We programmed our joint likelihood in the templated C++ code structure required to conduct inference with the TMB package [47] for the R computing environment [32].

## Results

In the U.S., we flew 15720 km and photographed 5830 km² of sea ice habitat during "on effort" portions of U.S. surveys (e.g., excluding thick fog or banking turns) between April 7 and May 31, 2016. Mean "on effort" flight length was 629 km (range 45-1293 km), in flights that averaged 4.4 hrs each (range 2.6-5.5 hr). Although the U.S. survey crew abstained from flying in extremely poor weather conditions, flying in marginal conditions with patchy fog often resulted in relatively long flights (e.g., 2.6 hrs) that yielded relatively little data (e.g., 45 km of "on effort" flight tracks) (Fig 1, S1 Video). We detected three groups of polar bears in U.S. waters using our thermal detection algorithm, and visually detected another five groups (Fig 1).

The Russian survey team largely limited flights to pre-planned routes, selecting days to fly based on good flight conditions. After removing one transect in which persistent low lying fog precluded consistent observations, the thermal swath of the Russian infrared scanner covered

$\approx 5414 \text{km}^2$ in seven flights totalling 11604 km. Russian survey flights detected five groups of bears using thermal sensors, and another 44 groups using a combination of human visual detections and manual searches of photographs.

Mean group size of polar bears in U.S. surveys was 1.25 (SE 0.25) and 1.39 (SE 0.10) in Russian surveys. For mothers with dependent young, the mean number of observed dependents was 1.4 (SE 0.13).

In U.S. thermal detection trials, our thermal detection algorithm correctly identified 8 of 12 bear groups for a detection probability of $\hat{p} = 0.67$. The Malachite-M thermal sensor used in Russian surveys did not detect any bears when temperatures were $< -5°C$; even in warmer temperatures it only detected 1 of 9 bear groups that were seen by human observers and deemed to be within the thermal detection swath.

Our joint models for polar bear abundance and tracks produced abundance estimates ranging from $\hat{N}^* = 3,435$ (95% CI: 2,300-5,131) to $\hat{N}^* = 5,444$ (95% CI: 3,636-8,152) depending on the proportion of bears assumed to be missed on the transect line during Russian surveys (Table 1). Our models fit polar bear counts well, but considerable lack-of-fit was discernible in models for bear tracks (Fig 3). Such lack-of-fit was likely an artifact of trying to join different types of observations (on the U.S. side: systematic sampling of photographs; on the Russian side: subjective determinations of the number of distinct tracks made by human observers). Although the spatial distribution of polar bear tracks looked similar to estimated bear densities (Fig 4), it did not help explain bear abundance better than other physiographic variables. In particular, after accounting for effects of other covariates (Fig 5), confidence intervals for track effect estimates considerably overlapped zero (e.g., $\alpha_{\text{tracks}} = 0.19$, SE 0.67). However, the correlation between the estimated track surfaces and estimated polar bear density surfaces was $\rho = 0.63$, higher than any of the other individual covariates ($\rho = -0.45$ for `dist_land`; $\rho = -0.45$ for `northing`; $\rho = -0.40$ for `easting`; $\rho = 0.25$ for `RSF`; $\rho = 0.15$ for `ice`).

**Table 1. Estimates of polar bear abundance for different regions and assumptions about *g*(0) in Russian survey flights, along with 95% log-based confidence intervals.** Regions include the full study grid ('Chukchi'), mean (i.e., time-averaged) abundance for those cells of the survey grid with centroids in U.S. waters ('U.S.'), mean abundance in Russian waters ('Russia'), mean abundance in the portions of our study area that overlapped the polar bear specialist group boundary ('PBSG'), and an estimate for the area used for capture and release of bears in a mark-recapture study of polar bears in the Chukchi Sea ('Regehr' [8]). For a map of the study area and related regions, see Fig 1).

| Region | *g*(0) | $\hat{N}^*$ | 95% CI |
|---|---|---|---|
| Chukchi | 1.0 | 3435 | (2300-5131) |
| Chukchi | 0.8 | 4196 | (2807-6273) |
| Chukchi | 0.6 | 5444 | (3636-8152) |
| U.S. | 1.0 | 340 | (144-801) |
| U.S. | 0.8 | 362 | (155-848) |
| U.S. | 0.6 | 393 | (169-914) |
| Russia | 1.0 | 3095 | (2054-4666) |
| Russia | 0.8 | 3834 | (2541-5784) |
| Russia | 0.6 | 5051 | (3344-7629) |
| PBSG | 1.0 | 3104 | (2099-4590) |
| PBSG | 0.8 | 3798 | (2565-5623) |
| PBSG | 0.6 | 4936 | (3327-7322) |
| Regehr | 1.0 | 126 | (52-306) |
| Regehr | 0.8 | 133 | (55-323) |
| Regehr | 0.6 | 143 | (59-347) |

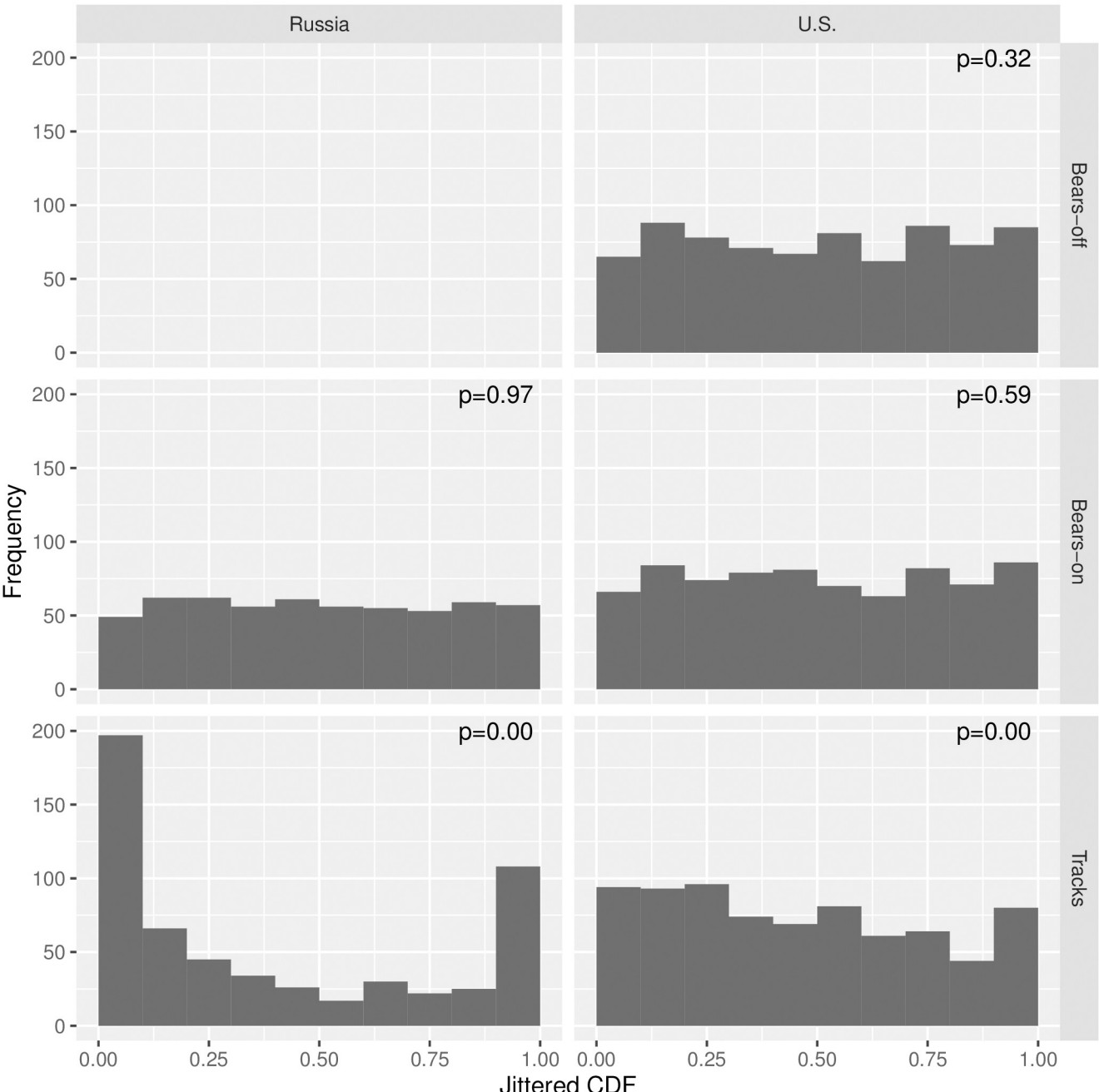

**Fig 3. Goodness-of-fit diagnostics.** Randomized quantile residuals (RQRs) for assessing goodness-of-fit for models fit to polar bear encounter data. RQRs should be uniformly distributed on (0,1) for a well-fitting model. Also presented are $\chi^2$ test p-values to assess uniformity.

Distribution maps suggested higher abundance and track densities within several hundred kilometers from land when sea ice is present (Fig 4), and relatively low densities far out on the pack ice. Our models predicted polar bear abundance to be substantially (close to ten times) higher in Russian than U.S. waters (Table 1). Point estimates of abundance in the Regehr et al. (2018) intensive capture-recapture study area were 126-133 depending upon $g(0)$ assumption

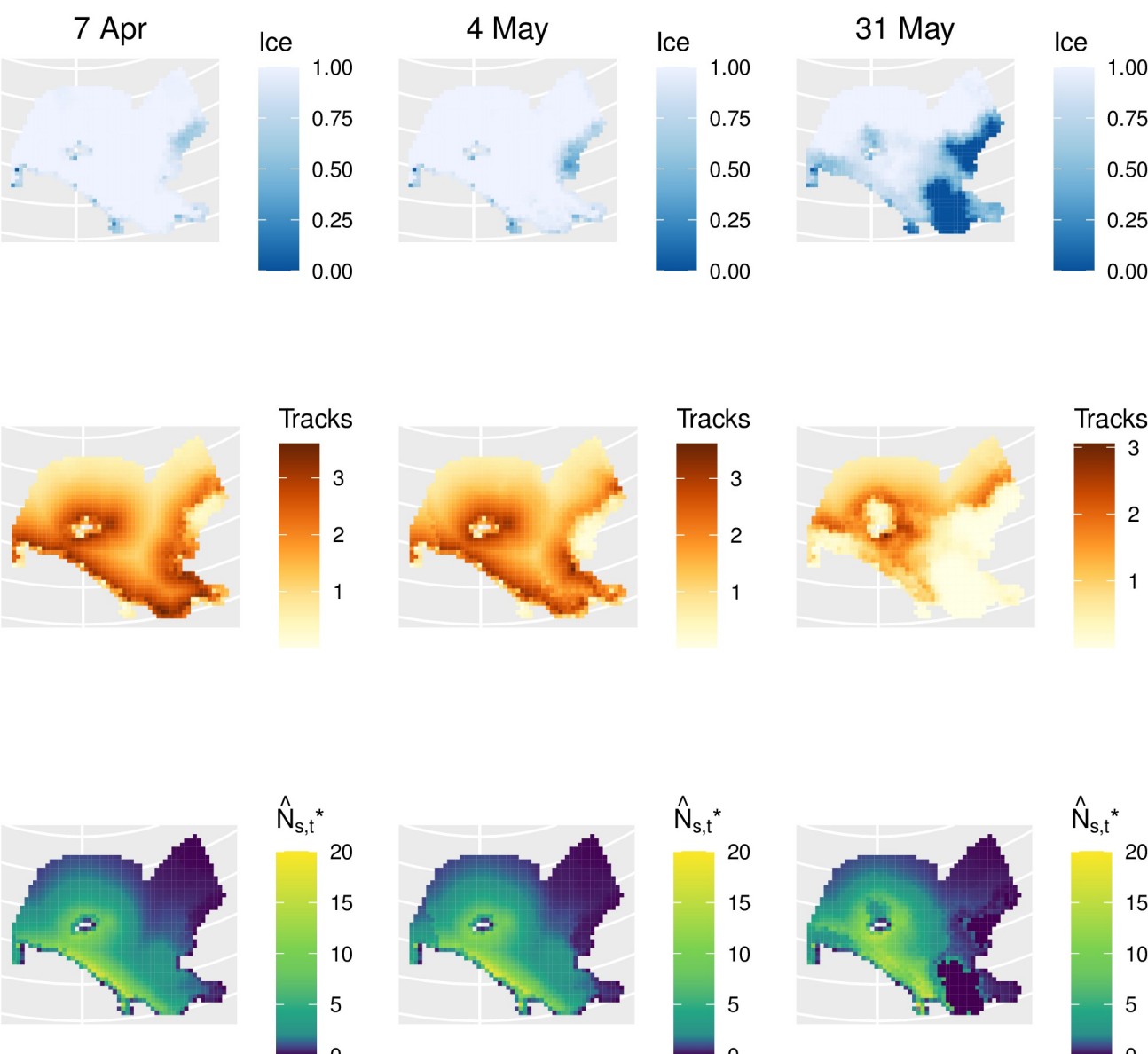

**Fig 4. Spatio-temporal maps of sea ice and predicted polar bear distribution.** Remotely sensed sea ice concentration values (top row), estimated polar bear track index (middle row), and predictions of polar bear abundance at the beginning, middle, and end of 2016 aerial surveys of the eastern Chukchi Sea ($g(0) = 0.8$ scenario). The polar bear track index is an estimate of the proportion of photographs that would contain polar bear tracks had photographs been taken in all grid cells and on all days of the survey. Predicted abundance is calculated as $\hat{N}_{s,t} = \hat{N}\hat{\pi}_{s,t}\mu_g$. Note that the scale of shading on abundance plots is nonlinear (i.e., low densities are visible as light blue and teal colors).

(Table 1). By contrast, an integrated population model produced a posterior mean of 78 (95% credible interval 36-138) bears in this region [8], which was within the confidence intervals of our predictions.

Translation of polar bear abundance to density is somewhat complicated by the fact that sea ice changed considerably throughout the survey (S1 Video). Calculating the area of sea ice habitat as $\text{ice}_t = \sum_s \text{ice}_{st}$ for time $t$ and spatial grid cell $s$ leads to values ranging from 824,000 km$^2$ at the beginning of the study to 610,000 km$^2$ at the end of the study. Conservative point estimates

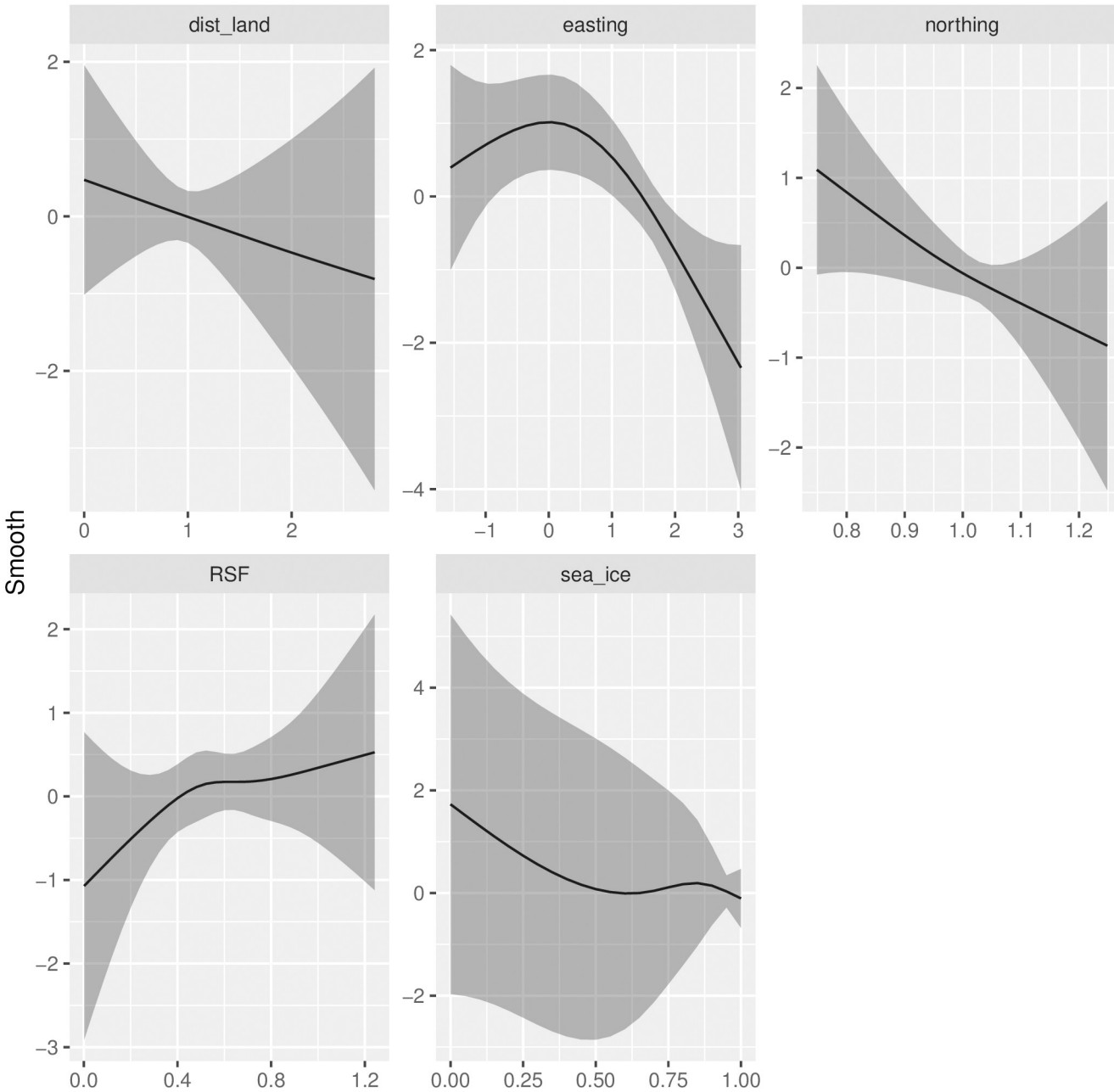

**Fig 5. Covariate effects.** Estimated smooth effects of covariates on polar bear abundance (black line), together with 95% confidence intervals (grey shading). Note that distance from land, easting, and northing effects were standardized to have a mean of 1.0 prior to analysis. Polar bear tracks were modeled as a simple linear effect on abundance so do not appear here.

of densities (i.e., for $g(0) = 1.0$) therefore ranged from 0.004 bears/km$^2$ of sea ice habitat at the beginning of the study to 0.006 bears/km$^2$ at the end of the study. Making this same calculation with respect to international boundaries results in conservative point estimates of 0.011— 0.020 bears/km$^2$ of sea ice in Russian waters and 0.001-0.002 bears/km$^2$ in U.S. waters during spring of 2016.

## Discussion

Our study represents the first comprehensive attempt at using aerial surveys to estimate polar bear abundance in the Chukchi Sea region. Polar bears are difficult to enumerate because they occur at low densities over large geographical areas with few human settlements from which to execute surveys. Long-term (i.e., approximately 35 years, which corresponds to three polar bear generations [50]) population trends are only available for 7 of 19 subpopulations worldwide [51]. Previous estimates of polar bear abundance have primarily been based on intensive physical or genetic capture-recapture surveys conducted over multiple years [52–55], or on autumn distance sampling surveys using helicopters in regions where polar bears occur at high densities on land [9, 10, 17]. Instead, our surveys were conducted with fixed wing aircraft at faster speeds, higher altitudes, and with greater fuel capacity. Conducting instrument-based surveys in the spring provided a permanent record of fine-scale sea-ice habitat and data on distribution and abundance of key prey species such as ringed (*Pusa hispida*) and bearded (*Erignathus barbatus*) seals. Such surveys (ideally using automated LWIR detection) are therefore an attractive option, given the potential for increased efficiency, longer ranges, reduced disturbance to bears, increased safety of survey personnel, and comprehensive data collection. At the same time, it is important to note that bear densities, as well as sample sizes, will be lower in the spring relative to late summer or fall surveys when the quantity of polar bear habitat (and therefore the size of potential study areas) is reduced [56].

Our intention was to base estimation on infrared detections made throughout the surveyed area. However, the thermal sensors employed in U.S. and Russian surveys performed differently. The algorithm applied to thermal imagery collected during U.S. surveys detected 67% of polar bear groups in experimental trials, whereas the Malachite-M sensor used in the Russian survey never detected bears when ambient temperature was below −5˚C; even above −5˚C, very few bears were registered in the IR imagery compared to other modes of detection (e.g., visual detections by crew, post hoc manual examination of photographs). As such, we used distance sampling methods to model detection of bears in Russian surveys. This approach requires assumptions about the proportion of bears detected on the transect line ($g(0)$), for which empirical estimates were unavailable.

Our model-based estimates of abundance varied considerably based on the $g(0)$ value assumed for Russian surveys. For example, under the optimistic scenario that 100% of bears were detected on the transect line ($g(0) = 1.0$), total estimated abundance was 3,435 (95% CI: 2,300-5,131) for an area bounded by Chaunskaya Bay, Russia, to the west and Point Barrow, Alaska to the east (Fig 1). We suggest this confidence interval represents a minimum plausible estimate given that lower values of $g(0)$ will always produce higher abundance estimates. For example, use of $g(0) = 0.6$ produced an estimate of 5,444 (95% CI: 3,636-8,152). Although these values vary considerably, they are of similar magnitude to that produced by a recent integrated population model fitted to data from a 2018-2016 live-capture study on sea ice west of Kotzebue, Alaska, USA, and extrapolated to other portions of the CS subpopulation boundary ($\hat{N} = 2,937$; 95% CI 1,522-5,944) [8]. The integrated population model was presumably most reliable in the area where physical marking and recapture events took place. Aerial survey estimates in this smaller region (Fig 1, Table 1) were similar to (though slightly larger than) those estimated from the integrated population model (78 bears [8]). Considering changes in sea-ice area during the survey, and conservatively assuming $g(0) = 1.0$, estimates of absolute density ranged from 0.004–0.006 bears/km$^2$ in our combined study area (U.S. + Russia). These values are similar to a mean estimate of 0.0036 (95% credible interval = 0.0019–0.0073) for the period 2008–2016 within the CS subpopulation boundary, based on an integrated population model with density extrapolation [8]. Estimated densities of bears over U.S. waters from our analysis

($\approx$0.001 bears/km$^2$) were lower than estimated from springtime aerial surveys in the northern Bering, eastern Chukchi, and western Beaufort seas conducted in 1987 (0.002 bears/km$^2$; [56]), and for late summer and fall surveys in these regions (0.005-0.007 bears/km$^2$; [56, 57]).

Given the impact of $g(0)$ on abundance estimation, it is important to review available data that might inform a likely range of values. Multiple-observer polar bear surveys reported in the literature have typically resulted in high $g(0)$ values (e.g., 0.85—0.90), although these estimates are mostly based on helicopters flying at lower speeds, lower altitudes, and over land (S2 Appendix). Detection rates from these studies were thus likely to be higher than those for in situ observations made during Russian survey flights because helicopter observers are closer to animals and have increased visual contrast compared to observations made fixed wing aircraft flying over sea ice. However, in addition to human detections made from the air (which would be expected to have lower $g(0)$ values than reported in the literature), we also detected a relatively large number of bears through post hoc visual inspection of photographs and incorporated these into our distance sampling analysis. Conditions being equal, this process should increase $g(0)$ relative to multiple observer surveys. Owing to the differences in survey protocols used here versus reported in the literature, we are currently unable to provide advice on which value of $g(0)$ is most appropriate, and suggest it be a topic for future research.

Our analysis incorporated covariates such as sea ice, RSFs, and polar bear track indices as predictors of polar bear density within a generalized additive modeling framework. Even though none of the covariates had strong correlations with estimated density by themselves, the combined suite predicted polar bear counts well (Fig 3). Development of track indices required their own modeling exercise; however, we propagated uncertainty in the estimated index into the analysis for bear abundance through a joint likelihood modeling framework. To our knowledge, this is the first attempt to use tracks to predict polar bear abundance. However, it is important to note that polar bear tracks indicate presence of bears at some point in time, but the longevity, visibility, and location of tracks was likely a function of many factors including sea-ice dynamics, surface characteristics (e.g., hard ice vs. deep snow), and weather. Although further investigation with more consistent track sampling protocols would be worthwhile, the lack of support for a track density effect as a predictor of polar bear abundance suggests that their utility for population monitoring (i.e., as a relative abundance index) may be limited. On the other hand, our estimated track index did have the highest correlation with estimated abundance among predictive covariates ($\rho = 0.63$). Interestingly, RSFs had a low correlation with estimated polar bear abundance ($\rho = 0.25$). Comparing maps of RSFs (S1 Appendix) to abundance maps, it is evident that RSFs do a poor job of predicting the high densities of polar bears we observed in Russian waters south of Wrangell Island. It is important to realize that these RSFs were developed based on a relatively small sample of non-denning adult females fitted with radiocollars [11], in a relatively small area near Kotzebue, Alaska, and may not represent habitat use of the larger subpopulation. Alternatively, there may be substantial interannual variability in polar bear distribution throughout the Chukchi Sea so that average space use does not always reliably predict relative densities in individual years.

Our experiences with instrument-based aerial surveys for polar bears led to several suggestions for future improvement. First, infrared sensors do not all reliably detect polar bears and the instrument used in Russian surveys should be replaced with a more sensitive model. Second, we suggest that for a given sensor or sensor array (e.g., IR and UV) a designed study should be performed to obtain a reliable estimate of detection probability for use in abundance estimation. In U.S. surveys, we conducted experimental flyovers of bears which yielded a serviceable estimate. However, in Russian surveys no independent estimate was available. Third, if future surveys use distance sampling methods, mark-recapture distance sampling with two observers [58–60] may provide an estimate of $g(0)$. However, such methods require statistical

independence of observations, at least at a single distance value (if point independence is assumed [59, 61]), which may be violated if both observers key into unique features of animals to aid detection (e.g., distinctiveness, movement). As such, an independent estimate of $g(0)$ derived from a different detection approach is highly desirable. For instance, if infrared detections are automated, the proportion of animals also detected by human observers provides an independent estimate of detection probability. Unfortunately, in our surveys, all forms of detection in Russian surveys were dependent (for instance, photographs were often triggered when human observers detected animals) and such "double sampling" estimates of detection probability would have been inappropriate. If double sampling is used in future surveys to estimate detection probability, some level of autonomy is needed between infrared detections, photographs, and human observers. Finally, we did not originally intend to collect data on polar bear tracks, which led to different protocols in U.S. and Russian surveys. Ideally, the same protocol (e.g., systematic sampling of automated photographs as used on the U.S. side) would be used to collect data on polar bear tracks. Although we used a spatial point process model to join these two data sources, the fit of this model was poor, particularly for Russian track data (Fig 3).

## Additional limitations and potential impact on estimates

Several other assumptions of our modeling approach deserve further discussion, especially as they relate to reliability of abundance estimates. For instance, we assumed that the total number of bears within our study area was constant as surveys were being conducted. Our modeling framework allowed bears to redistribute with changing conditions (e.g., as sea ice melted), but not to move in or out of the study area. Simulation studies have shown this assumption helps stabilize abundance estimates [40], and was crucial in this application given the low degrees of freedom. As such, our estimates are best interpreted as the average number of bears in the study area while sampling was being conducted.

We also assumed that bears selected for experimental flyovers were representative of the population with respect to infrared detectability. We were initially concerned that estimates of detection probability were biased high because detection trials were performed for a small number of animals visible to human observers (e.g., trial bears were usually moving and more often on flat ice than rubble). However, subsequent testing of infrared cameras (using captive bears and during additional flights in 2019) suggest that bears are similarly detectable in different behavioral states except after recently emerging from water (E. Moreland, unpublished data). Potential methods to improve automated detection include additional sensors (e.g., UV spectrum) and approaches to find and classify bears in multispectral imagery (e.g., artificial intelligence).

Our results are predicated on a relatively small sample size. First, although sample sizes were greater (49 polar bear groups in Russia, 8 on the US side) than for previous pilot aerial surveys in the Chukchi Sea [56, 57], they were smaller than would be ideal for fitting density surface models. Distance sampling texts (e.g. [42]) recommend 60-80 detections for estimating the parameters of a detection function, and we were somewhat under this mark. The fact that two different types of survey approaches needed to be combined within the same model only served to increase the number assumptions that needed to be made and the number of parameters that needed to be estimated. These elaborations decrease the overall reliability of our estimates relative to a survey with consistent technology and methodology throughout.

Finally, abundance estimates relied on estimated relationships between covariates and polar bear density, which can be problematic when extrapolated past the range of observed data [62, 63]. In our case, there were relatively few bears estimated to be in regions where we did not

sample (e.g., the north and west edges of our study area). For instance, a time-averaged estimate of polar bear abundance restricted to grid cells outside of the PBSG boundary was 331 bears (with the conservative $g(0) = 1.0$ scenario). Therefore, it does not seem that extrapolations near study area boundaries are driving the magnitude of our estimates. Nevertheless, this is something to be aware of when applying spatio-temporal models to survey data in other settings.

## Conclusion

Despite the methodological and technical challenges in this initial application, we are optimistic about the use of instrument-based aerial surveys for polar bears. Our approach has the potential to estimate distribution and abundance with minimal disturbance to bears and less risk to human safety relative to existing methods such as live-capture and genetic sampling using biopsy darts. The increased range of fixed-wing aircraft used in our surveys could increase access to very remote subpopulations where sample sizes from helicopter-based studies have been small (e.g., Kane Basin [64]). Furthermore, instrument-based surveys provide information on the distribution and abundance of ice-dependent seals, the primary prey of polar bears [65]. Recent studies have shown that relationships between sea-ice conditions and polar bear demography are variable in time and space [55, 66, 67], emphasizing that additional data on trophic relationships and ecosystem function are important to understanding the effects of climate change on polar bears. Refinement of future survey protocols, implementation of detection algorithms trained specifically on polar bears, and improved estimates of detection probability will increase the reliability and precision of abundance and distribution estimates derived from instrument-based aerial surveys.

## Supporting information

**S1 Appendix. Description of the development of polar bear resource selection layers for inclusion in model to estimate the number of polar bears in the Chukchi Sea polar bear subpopulation.**
(DOCX)

**S2 Appendix. Supplementary information on distance sampling.**
(DOCX)

**S1 Video.**
(MP4)

## Acknowledgments

We thank all researchers and technicians who participated in ChESS surveys, processed imagery, and organized data. In the U.S, this included G. Brady, S. Brown, M. Cameron, C. Christman, S. Dahle, S. Hardy, B. Hou, C. Johnson, E. Richmond, and A. Willoughby. In Russia, this included D. Glazov (Severtsov Institute) and Dr. D. Litovka (Government of Chukotka). We thank N. Chernook and V. Asyutenko (ANO Ecofactor) for image processing and Dr. N. Platonov (Severtsov Institute) for survey on-the-ground support. Dr. V. Burkanov (NPWC) provided overall supervision of the Russian survey planning and implementation. Views and conclusions in this article represent the views of the authors but do not necessarily represent findings or policy of the U.S. National Oceanic and Atmospheric Administration or U.S. Fish & Wildlife Service. Any use of trade, firm, or product names is for descriptive purposes only and does not imply endorsement by the U.S. Government.

## Author Contributions

**Conceptualization:** Paul B. Conn, Vladimir I. Chernook, Erin E. Moreland, Eric V. Regehr, Alexander N. Vasiliev, Stanislav E. Belikov, Peter L. Boveng.

**Data curation:** Vladimir I. Chernook, Erin E. Moreland, Irina S. Trukhanova, Alexander N. Vasiliev.

**Formal analysis:** Paul B. Conn.

**Funding acquisition:** Vladimir I. Chernook, Irina S. Trukhanova, Eric V. Regehr, Peter L. Boveng.

**Investigation:** Irina S. Trukhanova, Alexander N. Vasiliev.

**Methodology:** Paul B. Conn, Vladimir I. Chernook, Erin E. Moreland, Irina S. Trukhanova, Alexander N. Vasiliev, Ryan R. Wilson, Peter L. Boveng.

**Project administration:** Vladimir I. Chernook, Erin E. Moreland, Irina S. Trukhanova, Alexander N. Vasiliev, Peter L. Boveng.

**Resources:** Peter L. Boveng.

**Software:** Paul B. Conn, Alexander N. Vasiliev, Ryan R. Wilson.

**Supervision:** Vladimir I. Chernook, Peter L. Boveng.

**Validation:** Paul B. Conn.

**Visualization:** Paul B. Conn, Erin E. Moreland.

**Writing – original draft:** Paul B. Conn, Erin E. Moreland, Eric V. Regehr.

**Writing – review & editing:** Paul B. Conn, Vladimir I. Chernook, Erin E. Moreland, Irina S. Trukhanova, Eric V. Regehr, Alexander N. Vasiliev, Ryan R. Wilson, Stanislav E. Belikov, Peter L. Boveng.

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
