## [Decision Letter · Decision Letter 0]

2 Oct 2020

PONE-D-20-23811

Aerial survey estimates of polar bears and their tracks in the Chukchi Sea

PLOS ONE

Dear Dr. Conn,

Thank you for submitting your manuscript to PLOS ONE. After careful consideration, we feel that it has merit but does not fully meet PLOS ONE’s publication criteria as it currently stands. We invite you to submit a revised version of the manuscript that addresses the points raised during the review process. Reviewer 1 has make substantial and useful comments, which you will require to address in your revision (see detailed comments on the attached PDF file). The decision for major revision does not indicate an initial acceptance of your manuscript for publication. Your revised version will be peer-reviewed.

We look forward to receiving your revised manuscript.

Very best,

André Chiaradia, PhD

Academic Editor

PLOS ONE

Journal Requirements:

2.We note that [Figure(s) 1 and 3] in your submission contain [map/satellite] images which may be copyrighted. All PLOS content is published under the Creative Commons Attribution License (CC BY 4.0), which means that the manuscript, images, and Supporting Information files will be freely available online, and any third party is permitted to access, download, copy, distribute, and use these materials in any way, even commercially, with proper attribution. For these reasons, we cannot publish previously copyrighted maps or satellite images created using proprietary data, such as Google software (Google Maps, Street View, and Earth). For more information, see our copyright guidelines: http://journals.plos.org/plosone/s/licenses-and-copyright.

1.    You may seek permission from the original copyright holder of Figure(s) [1 and 3] to publish the content specifically under the CC BY 4.0 license. 

4.Thank you for providing the following Funding Statement: 

[Funding for surveys was provided primarily by the U.S. National Oceanic and Atmospheric Administration (NOAA) and the U.S. Fish & Wildlife Service (USFWS).  Support for Russian surveys was provided by NOAA through the North Pacific Wildlife Consulting, LLC (http://www.northpacificwildlife.com/). Portions of the analysis were supported by joint subaward NA17NMF4720289, project 1813, from the North Pacific Research Board and The Prince William Sound Oil Spill Recovery Institute (https://www.nprb.org/core-program/about-the-program/; PL, ER, IT, EM, and PC were principal or co-investigators).  Additional support for data processing and survey logistics on the Russian side was provided by USFWS, the RPO Marine Mammal Council (https://marmam.ru/en/) and WWF Russia (https://wwf.ru/en/about/) in funding agreements with VC.  External funders (e.g., NPRB, WWF Russia) had no role in study design, data collection and analysis, decision to publish, or preparation of the manuscript.].

We note that one or more of the authors is affiliated with the funding organization, indicating the funder may have had some role in the design, data collection, analysis or preparation of your manuscript for publication; in other words, the funder played an indirect role through the participation of the co-authors.

If the funding organization did not play a role in the study design, data collection and analysis, decision to publish, or preparation of the manuscript and only provided financial support in the form of authors' salaries and/or research materials, please review your statements relating to the author contributions, and ensure you have specifically and accurately indicated the role(s) that these authors had in your study in the Author Contributions section of the online submission form. Please make any necessary amendments directly within this section of the online submission form.  Please also update your Funding Statement to include the following statement: “The funder provided support in the form of salaries for authors [insert relevant initials], but did not have any additional role in the study design, data collection and analysis, decision to publish, or preparation of the manuscript. The specific roles of these authors are articulated in the ‘author contributions’ section.”

If the funding organization did have an additional role, please state and explain that role within your Funding Statement.

Please also provide an updated Competing Interests Statement declaring this commercial affiliation along with any other relevant declarations relating to employment, consultancy, patents, products in development, or marketed products, etc.  

Reviewers' comments:

Reviewer's Responses to Questions

**Comments to the Author**

1. Is the manuscript technically sound, and do the data support the conclusions?

Reviewer #1: Partly

2. Has the statistical analysis been performed appropriately and rigorously? 

Reviewer #1: I Don't Know

3. Have the authors made all data underlying the findings in their manuscript fully available?

Reviewer #1: Yes

4. Is the manuscript presented in an intelligible fashion and written in standard English?

Reviewer #1: Yes

5. Review Comments to the Author

Reviewer #1: Overall I feel this is an interesting paper, particularly creative in dealing with disjointed datasets and finding statistical procedures to overcome problems. I am not qualified to comment on the appropriateness of the details of the statistical procedures taken, though it seems to me the results are reasonable given the data collection/management situation.

Regardless of my inability to comment on the specifics of the stats, I found the paper to be quite confusing and though it's detailed and long I still found some information lacking. For example, why do the study in the spring in the first place? Surely there is a solid reason but it wasn't clear. Second, I'd prefer to see clear justification for the variables used in modeling; for example, a pet peeve of mine is using human-defined latitude and longitude information to describe animal populations, densities, behaviors, etc. Surely there is a more meaningful set of variables to use? Third, how was the statistical design of the transects made? Did it cover various types of habitat, when did you fly certain areas, etc.? I also think a clearer description of the study area would be good as well.

The various ways in which data were gathered, plus the complications, plus the two different countries make following exactly what you did - and why - difficult. Unfortunately I don't have a creative solution to help due to my still being fuzzy as to the details (sorry!).

My remaining comments are within the PDF but in general this seems to be an interesting statistical approach to dealing with a situation where lots of things didn't work as planned and the authors were able to work with that. However, it needs to be explained more plainly and clearly, I think, with the recognition that most readers won't know a lot of the terms you use or the biology of the bear, the climate of the region, etc. Don't take all that knowledge for granted!

6. PLOS authors have the option to publish the peer review history of their article (what does this mean?). If published, this will include your full peer review and any attached files.

Reviewer #1: No

---

## [Author Response · Author response to Decision Letter 0]

17 Nov 2020

Please see attached document for proper formatting. We are copying our response document in this field for completeness.

POINT-BY-POINT RESPONSES TO REVIEWER COMMENTS

Here we provide point-by-point responses to reviewer comments (our responses are in blue). We thank the reviewer for the time they took to examine our manuscript –it is evident that the reviewer spent a lot of time with our paper and we’re grateful for it. We think the following edits in responses to their comments will help clarify study design, survey conditions, and methodology, and make the paper more accessible to readers less familiar with polar bear biology.

Reviewer: 1

Reviewer #1: Overall I feel this is an interesting paper, particularly creative in dealing with disjointed datasets and finding statistical procedures to overcome problems. I am not qualified to comment on the appropriateness of the details of the statistical procedures taken, though it seems to me the results are reasonable given the data collection/management situation.

Yes, the disjointed nature of the data sets was challenging to deal with, but we’re happy with the product and are glad the reviewer was of a similar mindset.

Regardless of my inability to comment on the specifics of the stats, I found the paper to be quite confusing and though it's detailed and long I still found some information lacking. 

The level of detail and length was a function of how complicated the model was, and we feel is necessary so that readers would be able to duplicate our analysis. But we’ll make every effort to include information that is lacking.

For example, why do the study in the spring in the first place? Surely there is a solid reason but it wasn't clear. 

If this was entirely a polar bear survey, there would definitely be reasons to conduct the survey when sea ice has retreated and bears are more concentrated. However, this survey started off as a “seal” survey and was targeted at spring because this is the time of year when seals are molting and pupping and are thus most “available” to be counted. Doing the survey at this time of year will also allow us [in a future manuscript] to examine the spatial relationships between polar bear density and seal density (their primary prey). We agree that this could have been articulated a bit better, though. We have revised the second-to-last paragraph of the introduction to read

“Aerial surveys have been used to estimate polar bear subpopulations in a number of regions, and are typically conducted in late summer and early fall when there is less sea ice and bears are most concentrated (Aars et al. 2009, Stapleton et al. 2014, Obbard et al. 2015}. However, another possible approach is to conduct aerial surveys during the spring; although bears will be spread over a larger area, this approach allows one to study their distribution over the sea ice, and to simultaneously study the distribution of seal populations when they are engaged in pupping and molting and are therefore most available to be sampled. Conducting surveys in spring also allows for the potential of instrument-based approaches in which infrared cameras and coordinated digital color photography can be used to detect the warm bodies of animals on sea ice and confirm species identity.”

Second, I'd prefer to see clear justification for the variables used in modeling; for example, a pet peeve of mine is using human-defined latitude and longitude information to describe animal populations, densities, behaviors, etc. Surely there is a more meaningful set of variables to use?

We agree that using biologically meaningful covariates is best when good covariates are available. This helps both with ecological interpretation and (a bit more tenuously) with prediction (i.e., if covariates change, how would distributions be predicted to change). However, in our case, we are primarily interested in unbiased estimation of abundance. Use of covariates like latitude and longitude (or their 2-D projections), when applied in a flexible GAM-like modeling framework, can be used to fit spatio-temporal surfaces similar to those commonly employed in geostatistics and spatio-temporal statistics (see e.g. Wikle et al. 2019, section 4.5 @ https://spacetimewithr.org/). The advantage here is that one can account for spatial autocorrelation (clustering) of animals that can’t be accounted for with other covariates. If the goal is accurate estimation of animal distributions, we think this approach is best, particularly because none of our covariates (including RSF distributions that implicitly included other covariates like landfast ice proportions, ocean depth, and standard deviation of ice concentration) were especially effective at predicting survey counts. Given the magnitude of the different effect sizes (see Fig 4) it appears that easting and northing are some of the most impactful predictors here. 

For now, we have added the following paragraph to the “Explanatory covariates” subsection:

We included dist_land because seal densities (the primary prey of polar bears) are often highest close to land (Bengtson et al. 2005), and also because maternal dens are often located on land (Harington 1968) with high concentrations on Wrangel Island, Russia (Uspenski 1972) and along the northern Alaska coast (Durner et al. 2003). Since mothers and cubs emerge from dens in late winter and early spring (March-early April) we suspected there may be higher densities of bears along coastlines. We included ice since it has repeatedly been demonstrated to be an important determinant of polar bear habitat selection (Arthur et al. 1996, Durner et al. 2009, Wilson et al. 2014). Similarly, we included Water99 as a way to restrict polar bear use of habitat to those grid cells with >1% ice. Although bears can swim long distances, it was impossible to detect bears in the water and the proportion swimming at any one time is thought to be extremely low. The RSF distribution was a measure of habitat use developed from adult females; if habitat preferences of these bears mirror that of the population, we expected it would be a reasonable correlate for overall polar bear densities. Although the easting and northing covariates have little ecological meaning, we included them in models for polar bear counts because they enabled us to model coarse-grained spatial autocorrelation (clustering) in bear densities, as common in geostatistics and spatio-temporal statistical models (Wikle et al. 2019). Previous research (Wilson et al. 2014) found that polar bear resource selection can also depend on additional covariates such as proportion of landfast ice, ocean depth, variability of sea ice concentration, and average spring-fall chlorophyll concentration. Although we did not directly include these covariates in our models, most were implicitly included in our RSF covariate. For a description of which covariates were used in models for polar bear tracks and count data, see Models and model fitting, below.”

We have also updated our description of the RSF covariate with the statement

“Note that this covariate implicitly includes effects of landfast ice proportion, ocean depth, and standard deviation of ice concentration.”

Hopefully these changes adequately address the reviewer’s concerns; however, we are amenable to including additional covariates in our modeling efforts if the reviewer thinks it would be helpful.

Third, how was the statistical design of the transects made? Did it cover various types of habitat, when did you fly certain areas, etc.? 

We have added two additional paragraphs to the Aerial survey platform and protocols subsection (and modified a third) so that there is now much more detail on design considerations.

Aerial surveys of wildlife often use design-based statistical inference to estimate abundance. This approach requires survey planners to define a sampling frame of all possible transects, and to sample amongst those (often using systematic random sampling (Buckland et al. 2004) prior to conducting the survey. By contrast, model-based estimation, including modern density-surface models applied to data from line transect surveys (Miller et al. 2013) does not suffer from this requirement (though randomization can guard against subjective decisions that have potential to bias survey results through preferential sampling (Diggle et al. 2010, Conn et al. 2017). Model-based estimation has the key ramification that transect placement does not need to be allocated prior to the survey, permitting flexibility in decisions about when and where to survey, which is invaluable for modifying surveys when weather (often in our case, fog) precludes surveying in certain areas. 

A previous study examining alternative transect placement strategies for aerial surveys in the eastern Chukchi Sea (Conn et al. 2016) suggested reasonable precision and lack of bias when applying model-based estimation procedures to simulated polar bear count data. In that study, spreading effort out evenly over space resulted in slightly improved inference compared to stratified designs. This result was similar to what has been observed when fitting spatial models to environmental pollutant data: space-filling designs (in which sampling effort is spread evenly over space) tend to be optimal. Given this finding, our primary philosophy when making and altering flight plans (as sea ice conditions and weather changed, for instance) was to spread out sampling effort over time and space. We avoided surveying grid cells that were 100% open water, but otherwise attempted to structure transects to sample representative habitat within grid cells that did have ice.

U.S. and Russian survey protocols differed substantially, mostly owing to the constraints imposed by the survey platforms used. In the U.S., pre-survey flight planning supposed 27 flights with a mean range of 1293 km (range 1107-1365 km), with flights averaging 5.0 hrs each (range 4.3-5.3 hrs) centered on solar noon to maximize the number of seals that would be encounterd (Bengtson et al. 2005). However, variable weather conditions resulted in opportunistic survey effort and transects that varied considerably from these targets (see Results). The Russian survey team initially planned to fly 8 transects covering 13,000 km over 43 flight hours (roughly 1600 km and 5.4 hours per flight).

To avoid potential for bias due to preferential sampling (Diggle et al. 2010, Conn et al. 2017) crews of both aircraft were instructed to avoid fine scale targeting of ice habitat (e.g. following leads) or areas of high seal density when making and altering flight plans as sea ice and weather conditions changed. Owing to less flexibility in modifying transects while in flight, the Russian aircraft largely followed predetermined flight lines, while U.S. aircraft frequently made adjustments to sample areas that had not previously been sampled, or to avoid areas where visibility was poor (Fig 1).

This does not necessarily address the reviewer’s questions on when the different transects were flown. Our approach was to try to spread effort out over time (e.g. on the US side, fly out of Kotzebue, AK for a week, then fly transects out of Utqiaġvik (formerly Barrow), Alaska, and then head back to Barrow again). In this way, spatial and temporal effects on polar bear counts are less confounded. To help visualize where and when flights occurred, we produced a video that we now include as a supplementary file (“S1 Video”) that shows flight lines and observations for each day of the survey, superimposed on daily sea ice concentrations for the region.

I also think a clearer description of the study area would be good as well.

We have added in the sentence

“Our study area included all marine habitat within this region, including open water and areas covered by sea ice and open water (though we set polar bear abundance to zero in cells with no ice; see Models and model fitting.)” 

We also indicate that

“U.S. survey flights were conducted out of Kotzebue, Alaska, U.S.A. and Utqiaġvik, Alaska, U.S.A., whereas Russian flights were made from Pevek, Chukotka, Russia, and Provideniya, Chukotka, Russia.” 

We’ve also followed a number of suggestions that were made on the PDF mark up (see numbered responses below.

The various ways in which data were gathered, plus the complications, plus the two different countries make following exactly what you did - and why - difficult. Unfortunately I don't have a creative solution to help due to my still being fuzzy as to the details (sorry!).

My remaining comments are within the PDF but in general this seems to be an interesting statistical approach to dealing with a situation where lots of things didn't work as planned and the authors were able to work with that. However, it needs to be explained more plainly and clearly, I think, with the recognition that most readers won't know a lot of the terms you use or the biology of the bear, the climate of the region, etc. Don't take all that knowledge for granted! 

We believe we’ve now made things clearer while addressing remaining comments in the PDF. Here is a list of things of changes we’ve made in response to comments made directly on the PDF:

1) We changed “large ranges” in the abstract to “expansive, circumpolar distribution.” [the reviewer questioned whether we were referring to home ranges]

2) In response to the reviewer’s point about polar bear densities (“This depends on the time of year, age class, and location, doesn't it? And the question you're asking (with regard to accuracy and cost)? I'd like to see a little more information and updated data, rather than just referencing a single 25-year old paper to make your point.”), we now indicate

“For instance, previous estimates of springtime (April) density obtained from mark-recapture analysis ranged from 0.001-0.01 (mean 0.004) bears/km2 in the Canadian Arctic (Taylor and Lee 1995}, and 0.003 bears/ km2 for the Chukchi Sea (Regehr et al. 2018). Aerial survey estimates of polar bear densities are often conducted in late summer and early fall when polar bears are in higher concentrations because of reduced sea ice; densities at this time of year have ranged from 0.001 bears/km2 in the Barents Sea (Aars et al. 2009) where there is still substantial sea ice, to 0.02 bears/ km2 in Southern Hudson Bay when sea ice has largely receded and bears are confined to land (Obbard et al. 2015).” 

We are reluctant to conduct a completely thorough review of previous studies at this point, but hopefully this addition gives the reader a sense of how densities can vary by time of year and for different subpopulations.

3) On line 36, in response to a comment asking us to be more specific about “data”, we now indicate that “Although our survey generated count data for multiple species (including seals) . . . ”

4) We appreciated the comment “How many aerial surveys, how long per day, how many people were involved? Were people who surveyed the same people who searched the images?” but think this information is better placed in other sections than Study area. We now include information about crew size for each aircraft, as well as the number and duration of survey days. There were considerable differences in actual survey effort compared to planned survey effort (particularly for US surveys), so this information is spread out between the Aerial survey platform and protocols subsection and Results. We also indicate that the people who searched images were sometimes the same, and sometimes different from, those who flew the surveys (this information is included in the Data and data processing section subsection).

5) Lines 47-49. With reference to the queries about projection and grid cell resolution, we now include 

“We chose this scale because it corresponded to the resolution of sea ice imagery downloaded from the National Snow & Ice Data Center (NSIDC; see Explanatory covariates, below) and for consistency with previous analyses of ice-associated seals in the Bering Sea (Conn et al. 2014).”

In response to the comment “Also, is there a reason detail about the actual study area was left out? Average temperatures, sea ice conditions, wind, etc. so the reader can assess what you had to work in? I say this because later you mention the temperatures the cameras work in, OK so what temperatures are typical for the area at this time?” 

The only reason that this information was left out was for brevity, but we now provide information about temperatures, sea ice conditions, and wind in the form of an additional paragraph to the survey area section. We also show sea ice as a function of survey date now in a supplementary video (“S1 Video”). 

The comment “Why did you choose this time of year for your surveys? I think it would be good to spend a few more sentences prior to here talking about why spring vs. summer or fall when detections would possibly be easier (or maybe not?). If it's because of wanting to learn more about bears and sea ice in particular, say that more clearly. Trying to detect white animals on white ice seems pretty risky, so make more clear that this is either a pilot project to see if the tech works, or clarify why you needed to do the work in the spring time (most readers won't know this!).” was addressed previously in response to one of the general comments. [we included an additional paragraph in the introduction to address survey timing]

6) How did you design the survey transects and why? 

We addressed this in the general comments section. The short answer is we added two additional paragraphs to this section to address survey design.

7) Can you provide examples of what photos look like from each of these cameras?

We now include an extra figure showing what polar bears look like on infrared and color images (see what is now Figure 2).

8) Was fps defined earlier? Maybe frames per second is obvious to readers, but if not, please define. Now defined.

9) Ok so more polar bear tracks would likely correlate with greater polar bear density, I think that's what you're saying but why would tracks help explain the spatial variation in density? Wouldn't there be some other underlying physical (e.g., sea ice condition easier to traverse) or biological (near ice floe edge) reason for spatial variation? 

Perhaps this was stated poorly on our part. Because we expected tracks to correlate with density, our hope was using tracks would be a useful predictor of abundance (in the sense of increasing R2 of a standard regression analysis). But it isn’t really a causal mechanism, more of a correlative one. We’ve revised this to describe tracks as “. . . a potential correlate for polar bear density.” Although the reviewer’s other physical/biological covariates may indeed be better reasons for spatial variation, they are difficult to quantify as explanatory covariates from e.g. remotely sensed sea ice concentration rasters. This is a key point; we need to be able to have values for covariates for each grid cell for use in prediction. How to quantify, e.g. ice rugosity or ice floe edge density is not very straightforward.

10) My preference would be to justify a little more clearly the reasons for each of the variables. Why is distance to land important to polar bears? Resource selection function, etc.?

We now include a (rather long) additional paragraph summarizing our reasoning for including these covariates, and one additional paragraph talking about covariates that we did not use.

11) Ok but what characteristics of the sea ice? Presence, concentration, thickness (if that's even available)?

We stuck with concentration, as well as presence (through the Water99 covariate), as it has been the dominant covariate used in habitat selection analysis. Although sea ice products are increasing in sophistication, we are not aware of a reliable ice thickness product at present time.

12) What would easting and northing help you explain about the bear's biology and detection probability?

I see folks use lat or long as a covariate often; and often, when asked about why its being considered as a covariate, there is a more ecologically-relevant variable that isn't being considered but should be... bears (animals in general) don't know or care about lat/long no matter how we record it... so I have to ask here, what is it you're actually interested in knowing? In other words, what are easting and northing proxies for in the polar bear world? Temperature? 

Wind? (Is it colder or windier in certain places than others); Sea ice concentration, ocean temperature, etc.?

As we noted in our response to some of the ‘general comments,’ we view easting and northing (and more specifically, a GAM-type smooth representation) of these covariates as allowing us to account for spatio-temporal autocorrelation not explainable by other covariates. In a perfect world, animal distributions would be explainable entirely by causal predictors, but rarely in ecology are we so lucky. These covariates serve to account for areas of high (and low) abundance that are not predicted given the other covariates. We’ve included our reasoning in a large new paragraph justifying covariate inclusion.

13) I assume you kept distance to land, then, and got rid of northing?

Actually, we retained both, which should be evident in the Models and model fitting section. This information was just provided as a summary.

14) With reference to polar bear tracks, Why was the hope to derive a single covariate, rather than allow for mulitple covariates?

The issue is that if we developed one “track” covariate for the U.S. and another for the Russian part, there would be a discontinuity at the border; both with regard to the track model, and also with regard to estimates of polar bear density (in models where the discontinuous track covariate was used as a correlative predictor of polar bear densities). Since both likely vary smoothly (why would there be a country effect?), we preferred to try to join the two types of track observations with a single, underlying model for the density of tracks. 

15) If you don't have detection probabilities, strip widths (and thus, area) how can you say anything other than that you have a minimum count of bears?

Wouldn't you be technically limited in saying that you have only presence data for a given 25 x 25 km cell?

The reviewer is correct that we don’t have a detection probability or strip width for auxiliary bear counts, but that doesn’t mean we can’t come up with a way to model those counts (especially since we *do* have detection probability and strip width for IR counts). In this case, we’re estimating a scalar parameter, ξ, that relates auxiliary bear counts to the IR bear counts. In our case, we had 3 sightings on effort with a known detection probability, and 5 additional counts where bears were visible from the plane but out of the IR strip. Not surprisingly, the estimate of ξ comes out to 5/3. So in general what this says is that if we expect to get a count of C in a grid cell from the IR, we would expect, on average, to get a count of U=5/3*C auxiliary sightings. Please let us know if this explanation makes sense, or if we should add an explanation in the ms.

16) Were all detections in a single habitat type and did every grid with that habitat type have a bear detection?

We’re not quite sure what the reviewer is getting at with this question. There were very few grid cells that had observations, so clearly not every grid cell with the same habitat type had a bear detected. Four out of the five “auxiliary” detections occurred in habitat with leads close to the Bering Strait, while one was up closer to Utqiaġvik with more solid ice coverage (Fig 1). To our mind, it was useful to include these observations because it provided the model with the information that there *are* bears in the southeast portion of the study area.

17) Over how many search hours? How much time elapsed from the time of the first survey to the time of the last survey? How many transects? Please provide more detail.

We now provide information on the number of transects and number of search hours. The time frame (April 7 – May 31) was previously stated in the Methods section, but we restate it here as well now.

18) I still don't understand how you can calculate density if you don't have a way of knowing the total area over which you flew?

Well, we do know the total area over which we flew (for the IR in the US, and for the photo/visual distance sampling in Russia), and have a detection probability for both. Is the reviewer referring to the “Auxiliary counts” of bears on the US side? We’re hoping that we sufficiently clarified this in response to (15) above; but please let us know if there is still something that doesn’t make sense.

19) Why weren't these variables included in the modeling? If you gathered fine-scale habitat and prey data, wouldn't that help describe presence and density of bears?

There are several issues here. First, in order to do this type of Chukchi-wide population modeling, covariates need to be available for each grid cell in the study area. To use seal data in this manner, we would thus need to fit a similar model to seal counts to come up with seal population density maps. This is in the works, but not completed, and we eventually intend to try to relate polar bear and seal distributions in a subsequent paper. But the story is not quite as clear as one would like. For instance, ringed seal abundance is by far the highest in the land-fast ice of Kotzebue sound, where there are no polar bears (whether ringed seals use that area because it is a refugia from bears or for some other reason is not fully understood). So using seal density as a predictor of polar bear abundance is not as useful as one might think. 

Eventually it would also be nice to conduct modeling efforts with fine scale habitat data – e.g. extrapolating from in situ observations to sea ice conditions over the whole Chukchi Sea (perhaps using different ice type categorizations) but this type of modeling has not been conducted yet and would require collaboration with oceanographers and others who study sea ice.

20) Scientific names? Added

21) I think this is besides the point, the two different methods of gathering data tell us different things. As an aside, there seems to be a lot of worry among field biologists that "remote" methods of understanding populations like this would somehow replace MR studies and I think that's ridiculous. The sentiment here sounds like you're trying to ward off that notion/criticism and I don't think you need to.

We removed this sentence from the manuscript.

22) Regarding mention of GAM-like modeling framework in Discussion, the reviewer wrote ‘I think there could be more information or clarity about this in the methods section, it's not totally clear how this was employed.’

We were simply alluding to the smooth effects of covariates implemented using splines. We now state in the Models and model fitting section that “Models for latent polar bear abundance were considerably more complex, as we wanted the data to ``speak for themselves" by allowing smooth effects of covariates similar to the generalized additive modeling (GAM) framework (Wood 2017) commonly employed in modern density surface modeling of species distributions (Miller et al. 2013).”

23) Regarding poor fit of the track model: “can you remind what "poor fit" means here? We now refer to Fig. 3 here. These plots should be approximately uniform for a model that fits the data well; the low p-values and under- and over-predictions particularly for the Russian track data are indicative of lack-of-fit). 

24) Figure needs scale bar at a minimum, and a legend preferably. 

We added a scale bar and a north arrow to Fig. 1

---

## [Decision Letter · Decision Letter 1]

2 Feb 2021

PONE-D-20-23811R1

Aerial survey estimates of polar bears and their tracks in the Chukchi Sea

PLOS ONE

Dear Dr. Conn,

Thank you for submitting your manuscript to PLOS ONE. Also, thanks for your patience as the decision has taken longer than usual. We have received a new reviewer’s report on your revision 1. Like the report on your original submission, the reviewers feel your manuscript makes a useful contribution but needs more caution on the analysis and interpretation. I agree with them while aware of the difficulty to collect data on this species. However, the dataset and its treatment are still an issue that needs to be addressed. I have been pondering between an open rejection or major revision. I have decided for a major revision if you can address the reviewer’s concerns by revising your analysis and toning down your conclusions. If you go ahead with further review, your revision number 2 will be sent to peer-review. In summary, we feel that your manuscript has merit but does not fully meet PLOS ONE’s publication criteria as it currently stands. We invite you to submit a revised version of the manuscript that addresses the review process concerns.

We look forward to receiving your revised manuscript.

Very best,

Assoc Prof André Chiaradia

Academic Editor

PLOS ONE

Reviewers' comments:

Reviewer's Responses to Questions

**Comments to the Author**

1. If the authors have adequately addressed your comments raised in a previous round of review and you feel that this manuscript is now acceptable for publication, you may indicate that here to bypass the “Comments to the Author” section, enter your conflict of interest statement in the “Confidential to Editor” section, and submit your "Accept" recommendation.

Reviewer #2: All comments have been addressed

2. Is the manuscript technically sound, and do the data support the conclusions?

Reviewer #2: Partly

3. Has the statistical analysis been performed appropriately and rigorously? 

Reviewer #2: Yes

4. Have the authors made all data underlying the findings in their manuscript fully available?

Reviewer #2: Yes

5. Is the manuscript presented in an intelligible fashion and written in standard English?

Reviewer #2: Yes

6. Review Comments to the Author

Reviewer #2: This is an interesting attempt to estimate polar bear abundance in the Chukchi Sea. The major constraint on the study is the very small sample size. Only 8 groups were observed in US waters and 49 groups in Russian waters. By most standards, analyzing such a small sample would be of questionable merit given that 2 different platforms, different observation methods used, and varying conditions / habitats. I put little credibility in the actual results of abundance but the approach taken is rigorous (it’s unfortunate that more sightings weren’t made). My major concern with the study is the determination of the area to which the density estimates were applied: this is virtually a non-issue for the manuscript and totally ignored. The beige lines in Figure 1, if I’ve understood correctly, show the area to which the density estimates were applied. I suspect the large estimate is associated in part with applying the density to northern areas that may be almost unused at this time of year. I doubt the northern areas of the study area have a similar density to those nearshore but that is simply speculation (based on polar bear ecology and marine productivity) but the RSF may improve that fit although I found the description of how the RSF was applied cryptic. If the RSF is for only the period of the study, OK, that helps but I didn’t see much discussion on this point and all I know is that it’s inside the GAM. I’m not overly convinced that tracks are a useful component – they are incredibly dependent on snow conditions and weather.

On balance, I don’t really put much credence in the numbers produced but the authors have made the best of weak data and try to put the incredibly expensive data to use. As such, the study warrants presentation in a peer-reviewed journal so that others can learn of the various problems of conducting such a survey.

I think, however, a more cautious statement of caveats would be useful. Most of the major limitations of the study (e.g., sample size, extrapolation of RSFs, tracks, area of application) are glossed over.

Abstract (no line #)

“are larger than” – I suggest adding “point estimates” – there is no statistical difference between the earlier and current estimates (although it’s almost impossible to have a difference given the large confidence intervals

it’s unclear why the lower bound is considered useful but I’m OK with leaving it in – I don’t believe, however, that it is very useful as everyone will use the point estimates

2 - binomial name

5 – what does “demographic status” mean? This is an odd bit of wording.

17-8 – superscripts missing

106 & 113 – SI units (km/h) for speed

366 – it’s to its

385 – it is unusual to have groups as the primary sampling unit. Subadults, adult males, and solitary females would make up >50% of the population.

499-400 “Translation of polar bear abundance to density is somewhat complicated by the fact that sea ice changed considerably throughout the survey” This is only one aspect. It is extremely likely that the density estimate does not apply over the whole area shown in Figure 1 (beige grid). That the density along the coast was similar to the north areas does not fit well for what is known about polar bears. Relying on Regehr et al. 2018 is of questionable merit as there is little basis for extrapolating those estimates to the whole population. In essence, a major peril in density extrapolation to obtain a population estimate is knowing how density varies over the area. Without such insight, the population could be increased or decreased to any size based on the area to which density is applied. The RSF may assist as a proxy for density and the results are unclear on how much influence this had on the population estimate.

The caveat of area and density variation over the study area is pretty much a non-issue in the manuscript. This needs attention. The lack of context for the density results appears to be a major oversight. Comparing only to a single paper in the same area is questionable.

7. PLOS authors have the option to publish the peer review history of their article (what does this mean?). If published, this will include your full peer review and any attached files.

Reviewer #2: No

---

## [Author Response · Author response to Decision Letter 1]

18 Mar 2021

Please see attached document for proper formatting. A text-only version of our responses to reviewer comments is copied here for completeness but color is needed for optimal reading.

POINT-BY-POINT RESPONSES TO REVIEWER COMMENTS

Here we provide point-by-point responses to all reviewer comments (our responses are in blue). We thank the reviewer for the time they took to examine our manuscript. We believe the changes we have made in response to these comments will clarify the modeling description and more accurately convey key uncertainties of our analysis. 

Reviewer #2: This is an interesting attempt to estimate polar bear abundance in the Chukchi Sea. The major constraint on the study is the very small sample size. Only 8 groups were observed in US waters and 49 groups in Russian waters. By most standards, analyzing such a small sample would be of questionable merit given that 2 different platforms, different observation methods used, and varying conditions / habitats. I put little credibility in the actual results of abundance but the approach taken is rigorous (it’s unfortunate that more sightings weren’t made).

We agree with the reviewer that the sample size obtained in these surveys was not ideal for estimating animal abundance. Distance sampling, for instance, often suggests 60-80 encounters to be able to estimate the detection function accurately (see e.g. https://workshops.distancesampling.org/online-course/lecturepdfs/Ch4/L4-3%20Sample%20Size.pdf); our study used 49 for the Russian surveys, which is slightly under this recommendation. However, we do note that there are many examples in the published literature that use smaller samples to make inference about polar bear density from aerial surveys. For example, Evans et al. (2003) used a sample of 25 polar bear groups to estimate Chukchi-Beaufort polar bear density; McDonald et al. (1999) conducted inference using 15 polar groups; and Wiig and Derocher (1999) conducted inference using 29 bears. The low sample size is mostly attributable to the low densities of polar bears in these areas and are typical of springtime on-ice densities in much of the Arctic. In fact, our total survey effort (27,327 km) was greater than all 9 published studies we reviewed that use distance sampling to study polar bear abundance (Aars et al. 2009, 2017; Dyck et al. 2017; Evans et al. 2003; McDonald et al. 1999; Obbard et al. 2015; Stapleton et al. 2014, 2016; Wiig and Derocher 1999 – see S2 Appendix for citations), where the range of transects were all between 3,756 and 20,975 km. Our point is that while small, our sample sizes on the Russian side were comparable (and sometimes greater than) those in the peer-reviewed literature and that our study represents an unprecedented amount of survey effort compared to other polar bear surveys. The 49 groups counted on the Russian side allowed reasonable estimation of a detection function. The 8 groups observed on the US side were certainly not ideal and would be too few to estimate a detection function, but the additional 12 fly-over trials generated a reasonable estimate of detection probability that was independent of the main survey effort. The reviewer’s final point about varying conditions (specifically, sea ice melt) probably did impact the quality of our estimates near the Bering Strait where most melt occurred, but this was not an area where abundance was estimated to be high; in many areas (especially where abundance was estimated to be the highest), the quantity of sea ice was roughly constant during the period where surveys were conducted.

We made several changes and additions to the manuscript to ensure these comments are thoroughly addressed. First, we added text to the abstract and discussion emphasizing that the two survey platforms and other considerations meant that the methods used to generate abundance estimates depended on multiple assumptions: 

“. . . a number of factors (e.g., equipment issues, differing platforms, low sample sizes, size of the study area relative to sampling effort) required us to make a number of assumptions to generate estimates . . .”

Regarding sample size, we added the following to the discussion:

“At the same time, it is important to note that bear densities, as well as sample sizes, will be reduced in the spring relative to late summer of fall surveys when the quantity of polar bear habitat (and therefore the size of potential study areas) is reduced (McDonald et al. 1999).” (Lines 529-532)

Finally, we included a new section in the discussion, “Additional limitations and potential impact on estimates” as a place to collect the many, varied caveats associated with our analysis, including sample size, changing habitat, and different platforms/observation methods.

My major concern with the study is the determination of the area to which the density estimates were applied: this is virtually a non-issue for the manuscript and totally ignored. The beige lines in Figure 1, if I’ve understood correctly, show the area to which the density estimates were applied. I suspect the large estimate is associated in part with applying the density to northern areas that may be almost unused at this time of year. I doubt the northern areas of the study area have a similar density to those nearshore but that is simply speculation (based on polar bear ecology and marine productivity) but the RSF may improve that fit although I found the description of how the RSF was applied cryptic. If the RSF is for only the period of the study, OK, that helps but I didn’t see much discussion on this point and all I know is that it’s inside the GAM. 

There are two points here, one concerning possible extrapolations in the northern part of our study area, and one about the use of RSFs (and lack of a detailed description thereof) in the paper.

First, the reviewer is correct that there is potential for extrapolation bias in areas that are not surveyed. However, since our estimates are spatially explicit (see last row of Fig 4), one can calculate what abundance estimates are for certain areas in order to assess whether extrapolations should be regarded as problematic. For instance, taking a look for the estimate for the PBSG specialist group boundary (blue shading in Fig 1 that also overlaps with the beige grid), we get a time-averaged estimate of 3104 bears, compared to the estimate of the “full grid” of 3435 bears; stated another way, the abundance estimated in the unshaded area (primarily the northern and western portions of the study area, encompassing 32% of the study area) was estimated to be 331 bears. When looking at RSF plots which we now include in a new S1 Appendix, there appears to be predicted use in the far north of the study area. Therefore, we do not find evidence extrapolation is driving the magnitude of abundance estimates. Rather, it is the large number of encounters made in the southern half of the study area, particularly on the Russian side, that is driving the estimate. 

Perhaps part of the confusion is that the maps of estimated abundance (Fig 4, bottom row) have a nonlinear color scaling such that predicted abundance has to get very close to zero to achieve a blue hue; this was a result of us wanting to show regions with low abundance (e.g. light blue) in addition to regions with high abundance (yellow). We have now included the following text in the figure caption, which should hopefully clarify this:

“Note that the scale of shading on abundance plots is nonlinear (i.e., very low densities are visible as light blue colors).” 

Nevertheless, we agree with the reviewer’s broader point that our estimates are partially dependent on extrapolating to unsurveyed locations, which is potentially problematic if density-covariate relationships are unreliable in certain areas (e.g., the northern part of our study area). This is now addressed in the new subsection in the Discussion, entitled “Additional limitations and potential impact on estimates.” 

Second, we agree with the reviewer that the RSF covariate development was cryptically described and that the manuscript would benefit from a fuller treatment. We have thus included a new appendix (S1 Appendix) that describes how the RSF was developed, and provides plots of predicted space use during our surveys. Briefly, RSFs were developed using data from multiple years of satellite telemetry records from adult female polar bears equipped with radiocollars. We then used the RSF, together with environmental covariate values from each day of our aerial surveys, to predict relative habitat utilization in each grid cell. As indicated in our cover letter, this revealed a data formatting error in how RSF distributions were entered into the original analysis. Previously, just 1 RSF surface was used instead of the full suite of daily RSF distributions. Correction of this error decreased abundance point estimates by about 3%. The plots in this appendix are useful for comparing RSF distributions to density predictions from our model, as well as to raw maps of polar bear sightings. In particular, it is evident that RSFs based on telemetered bears do not adequately predict the high densities of bears observed in Russian waters south of Wrangell Island. The low correlation between RSFs and our estimated density surface (ρ=0.25) suggests that the RSFs, which were based on dta from a sample of females that were collared near Kotzebue, Alaska may not accurately reflect spatial habitat use of the entire Chukchi subpopulation, at least in 2016. We have introduced additional text to the discussion to expand on this point.

I’m not overly convinced that tracks are a useful component – they are incredibly dependent on snow conditions and weather.

Given the susceptibility of tracks to environmental conditions and our findings that the track effect confidence interval considerably overlapped zero (line 488), we agree to a limited extent. However, we haven’t seen other studies that have tried to relate polar bear density to track density in the literature. If one could use track density to predict polar bear density it would be quite advantageous, since it is much easier to encounter tracks than it is bears themselves. As such, we treated it as a hypothesis worthy of investigation. To address the reviewer’s point and provide context to this, the discussion now states:

“Although further investigation with more consistent track sampling protocols would be worthwhile, the lack of support for a track density effect as a predictor of polar bear abundance suggests limited utility for population monitoring (i.e., as a relative abundance index). On the other hand, correlation between polar bear density estimates and track density was 0.63, higher than any other individual covariate.” (Lines 596-601)

We also indicate that

“. . . it is important to note that polar bear tracks indicate presence of bears at some point in time, but the longevity, visibility, and location of tracks was likely a function of many factors including sea-ice dynamics, surface characteristics (e.g., hard ice vs. deep snow), and weather.” (Lines 592-596)

On balance, I don’t really put much credence in the numbers produced but the authors have made the best of weak data and try to put the incredibly expensive data to use. As such, the study warrants presentation in a peer-reviewed journal so that others can learn of the various problems of conducting such a survey.

The reviewer is certainly correct that the data obtained during this study were imperfect, especially with equipment issues on the Russian side. We tried to do our best given the data available and are glad the reviewer sees value in this. We also agree with the reviewer that an important aspect of our paper is to communicate the difficulties encountered with our approach, such that method might be improved in the future. Given the difficulty and expense of studying ice seals and polar bears, and the fact that diminishing sea ice is making live-capture studies for polar bears more difficult, we anticipate that aerial survey methods for these species will see increased application in the future. 

I think, however, a more cautious statement of caveats would be useful. Most of the major limitations of the study (e.g., sample size, extrapolation of RSFs, tracks, area of application) are glossed over.

We think this is a good suggestion and appreciate the reviewer’s attention to these important details. As mentioned earlier, we now devote a new subsection of the discussion to ``Additional limitations and potential impact on estimates” where we talk more specifically about sample size, extrapolation, and area of application. We also allude to these issues in the abstract now, indicating that

“. . . a number of factors (e.g., equipment issues, differing platforms, low sample sizes, size of the study area relative to sampling effort) required us to make a number of assumptions to generate estimates . . .”

Abstract (no line #)

“are larger than” – I suggest adding “point estimates” – there is no statistical difference between the earlier and current estimates (although it’s almost impossible to have a difference given the large confidence intervals

Changed

it’s unclear why the lower bound is considered useful but I’m OK with leaving it in – I don’t believe, however, that it is very useful as everyone will use the point estimates

We believe that the credible interval of the abundance estimate using g(0) = 1 does indeed provide useful information on the likely lower bound of abundance in this region, and so have retained the original text.

2 - binomial name

Added

5 – what does “demographic status” mean? This is an odd bit of wording.

We agree that this was unclear. Changed to “population trends”

17-8 – superscripts missing

Fixed

106 & 113 – SI units (km/h) for speed

Changed

366 – it’s to its

Fixed

385 – it is unusual to have groups as the primary sampling unit. Subadults, adult males, and solitary females would make up >50% of the population.

Using groups as the sampling unit is standard in aerial survey / distance sampling literature. The reason is that detections are made of groups of bears rather than individual bears independently. Assuming that individual bears are detected independently would inflate sample size and artificially increase precision. As such, detections are often modeled at the group (or “cluster”) level and then a separate model for group size is used to expand to the total number of individuals in the population. We now cite Buckland et al. (2001) as support for this procedure. (line 386)

499-400 “Translation of polar bear abundance to density is somewhat complicated by the fact that sea ice changed considerably throughout the survey” This is only one aspect. It is extremely likely that the density estimate does not apply over the whole area shown in Figure 1 (beige grid). That the density along the coast was similar to the north areas does not fit well for what is known about polar bears. Relying on Regehr et al. 2018 is of questionable merit as there is little basis for extrapolating those estimates to the whole population. In essence, a major peril in density extrapolation to obtain a population estimate is knowing how density varies over the area. Without such insight, the population could be increased or decreased to any size based on the area to which density is applied. The RSF may assist as a proxy for density and the results are unclear on how much influence this had on the population estimate.

The caveat of area and density variation over the study area is pretty much a non-issue in the manuscript. This needs attention. The lack of context for the density results appears to be a major oversight. Comparing only to a single paper in the same area is questionable.

Please see previous responses to “general comments.” In particular, we think the reviewer may have been misinterpreting our density plots (Final row of Fig 4.) which show how estimated density varies over the landscape. These estimated densities are a direct result of the GAM-like modeling framework where polar bear counts are related to environmental covariates, and these covariates are used to predict a density surface of bears in the study area (which varies spatially as well as temporally). Nevertheless, we provide additional context and collect caveats of the estimation process in a new section, “Additional limitations and potential impact on estimates”

It is difficult to provide many other comparisons of density in the study area because of the general lack of studies that have quantified density in this region, particularly in Russian waters. However, we now make comparisons to two pilot aerial surveys conducted in part over the U.S. portion of the Chukchi Sea (Evans et al. 2003; McDonald et al. 1999) – see lines 564-568. We also provide additional text in the Discussion comparing RSF distributions to our estimated density surfaces (Lines 602-610).

---

## [Editor Report · Decision Letter 2]

21 Apr 2021

Aerial survey estimates of polar bears and their tracks in the Chukchi Sea

PONE-D-20-23811R2

Dear Dr. Conn,

Thanks for your thoughtful reply while addressing concerns raised during the review process. The small sample size is a significant issue, but your caution on limitations of the study gave a good balance to have this hard-to-get data published. Without further ado, we are pleased to inform you that your manuscript has been judged scientifically suitable for publication and will be formally accepted for publication once it meets all outstanding technical requirements.

Very best,

Assoc Prof André Chiaradia

Academic Editor

PLOS ONE
---

## [Editor Report · Acceptance letter]

26 Apr 2021

PONE-D-20-23811R2 

Aerial survey estimates of polar bears and their tracks in the Chukchi Sea 

Dear Dr. Conn:

I'm pleased to inform you that your manuscript has been deemed suitable for publication in PLOS ONE. Congratulations! Your manuscript is now with our production department. 

Kind regards, 

on behalf of

Assoc. Prof. André Chiaradia 

Academic Editor

PLOS ONE